# Evaluation of the CMCC global eddying ocean model for the Ocean Model Intercomparison Project (OMIP2)

Doroteaciro Iovino[1], Pier Giuseppe Fogli[1], Simona Masina[1]

[1] Ocean Modeling and Data Assimilation Division, Fondazione Centro Euro-Mediterraneo sui Cambiamenti Climatici - CMCC, Bologna, Italy

*Correspondence to*: Doroteaciro Iovino (dorotea.iovino@cmcc.it)

**Abstract.** This paper describes the global eddying ocean-sea ice simulation produced at the Euro-Mediterranean Center on Climate Change (CMCC) obtained following the experimental design of the Ocean Model Intercomparison Project phase 2 (OMIP2). The eddy-rich model (GLOB16) is based on the NEMOv3.6 framework, with a global horizontal resolution of 1/16° and 98 vertical levels, and was originally designed for an operational short-term ocean forecasting system. Here, it is driven by one multi-decadal cycle of the prescribed JRA55-do atmospheric reanalysis and runoff dataset in order to perform a long-term benchmarking experiment.

To assess the accuracy of simulated 3D ocean fields, and highlight the relative benefits of resolving mesoscale processes, the GLOB16 performances are evaluated via a selection of key climate metrics against observational datasets and two other NEMO configurations at lower resolutions: an eddy-permitting resolution (ORCA025) and a non-eddying resolution (ORCA1) designed to form the ocean-sea ice component of the fully coupled CMCC climate model.

The well-known biases in the low-resolution simulations are significantly improved in the high-resolution model. The evolution and spatial pattern of large-scale features (such as sea surface temperature biases and winter mixed layer structure) in GLOB16 are generally better reproduced, and the large-scale circulation is remarkably improved compared to the low-resolution oceans. We find that eddying resolution is an advantage in resolving the structure of western boundary currents, the overturning cells, and flow through key passages. GLOB16 might be an appropriate tool for ocean climate modeling efforts, even though the benefit of eddying resolution does not provide unambiguous advances for all ocean variables in all regions.

## 1 Introduction

Ocean-sea ice models are built for a variety of applications. They are used for ocean and ice forecasting on short timescales, but they are also incorporated in coupled climate and Earth system models for sub-seasonal to decadal predictions and climate projections. An accurate representation of the ocean dynamics within the climate system is crucial to understanding drivers of climate change and variability, and to determining the ocean-ice influence on atmospheric circulation and ecosystems.

Despite the ongoing increases in computer power and improvements in techniques, a major challenge in climate model design is the trade-offs between the level of model complexity, the length of simulations, the choice of ensemble size and the spatial resolution of different climate components. In the Coupled Model Intercomparison Project Phase 6 (CMIP6, Eyring et al. 2016), the typical grid spacing for the ocean component of coupled climate models is still 1°, although some models were prepared at 0.25° horizontal grid spacing. Both resolutions lack an explicit representation of ocean mesoscale dynamics in most of the global domain. Eddy-rich ocean models improve the climate state with more accurate estimates of heat transport, boundary currents and ocean dynamics in key straits (Griffies et al. 2015, Hewitt et al. 2016, Roberts et al. 2019). Simulations of the global ocean domain at this resolution still require significant computational resources, which limits the number and length of runs and the capacity to optimize the model setup. However, thanks to the ever-increasing processing and storage capabilities of the supercomputers, running global models capable of resolving mesoscale dynamics has become feasible for

climate simulations. It is now necessary to assess to what extent the enhanced resolution translates into an improved ocean state.

Within the CMIP6, the Ocean Model Intercomparison Project (OMIP; Griffies et al., 2016) was proposed to trace the origins and consequence of model biases in ocean-sea ice configurations. OMIP provides an experimental and diagnostic framework for evaluating, understanding, and improving ocean and sea ice (together with tracer and biogeochemical components) of climate and earth system models. The essential element behind the OMIP is a common set of atmospheric and runoff datasets for computing surface boundary fluxes to drive the ocean–sea ice models. The phase 2 of OMIP (OMIP2) is forced by the

JRA55-do atmospheric forcing (Tsujino et al. 2018) developed from the Japanese 55-year Reanalysis (Kobayashi et al. 2015), and an updated freshwater runoff dataset (Suzuki et al., 2017). Eleven CMIP-class global ocean-sea ice models at low-resolution (~1°) have been intercompared and evaluated in Tsujino et al. (2020), identifying many improvements in the simulated fields in transitioning from OMIP Phase 1 (forced by the CORE-II dataset, Large and Yeager, 2009) to OMIP2. For example, the OMIP2 sea surface temperature (SST) reproduces the observed global warming at the end of the last century, the

warming hiatus in the 2000s, and the accelerated warming thereafter, all absent in OMIP1; the seasonal and interannual variations in SST and sea surface height are also improved. Many of the remaining model biases are mainly due either to biases in the shared atmospheric forcing, or to poor representation of ocean–sea-ice physical processes, some of which are expected to be mitigated by refining horizontal and/or vertical resolutions. High-resolution OMIP-2 experiments, performed with global ocean-sea ice systems at eddy-rich resolution (order of 1/10°) are presented and compared by Chassignet et al. (2020) in order

to isolate the improvements of ocean–sea-ice response to JRA55-do by increasing horizontal grid resolution.

Under this framework, several modeling centers started to perform multi-resolution studies (e.g., Storkey et al. 2018, Adcroft et al. 2019, Kiss et al. 2020, Li et al. 2020). Following the same approach, CMCC uses a hierarchy of ocean-sea ice configurations, with the aim of providing a relatively robust assessment of how climate-relevant changes in ocean mean state and variability are associated with the grid enhancement from non-eddying (low-resolution), eddy-permitting (medium-

resolution) to eddy-rich (high-resolution) configurations in our ocean components. We run OMIP-like simulations with the three models driven by the same forcing dataset, and we compare them in order to identify possible climate-relevant improvements in the ocean response as model resolution increases. It is worth mentioning that the models do not differ only in the horizontal resolution and associated physical parameters since the high-resolution simulation was configured independently for distinct scientific applications and followed a specific development strategy. Our non-eddying experiment

(ORCA1, nominally 1° horizontal grid spacing) shown here is the one used in Tsujino et al. (2020), designed as a component of the CMCC climate model (Cherchi et al., 2019) and Earth system model (Lovato et al., 2022) for CMIP6; the eddy-permitting configuration (ORCA025, 0.25° horizontal grid spacing) shares the same numerical framework and was configured as a component of the CMCC climate model (e.g. Roberts et al., 2020, Meccia et al., 2021) used in the High Resolution Model Intercomparison Project (HighRes MIP, Haarsma et al. 2016). Our eddy-rich configuration (GLOB16, 0.0625° horizontal grid

spacing) is designed to be for the operational short-term ocean forecast (https://gofs.cmcc.it/) and reanalysis systems. It undergoes continuous updates and is now used in international projects for mesoscale process studies at global (Treguier et al., 2023) and regional (e.g. Manral et al., 2023, Wang et al., 2023) scales. Our OMIP2 simulation at high resolution was made available in 2020, so it was not included in the intercomparison by Chassignet et al. (2020). Unlike the lower-resolution runs, it is not shared through the Earth System Grid Federation (ESGF) data server.

Because of the large computation resources required to run long hindcast simulations with GLOB16, only one JRA55-do cycle (61 years from 1958 to 2018) is analyzed in this paper - versus six JRA55 cycles for the low- and medium-resolutions.

This study aims to contribute to assessing how mesoscale processes affect the ocean spatial and temporal variability by comparing GLOB16 to the other two configurations, and to quantify the general improvement of many ocean model metrics by evaluating GLOB16 against observation-based estimates. The different configurations have not been developed

simultaneously and with no similar scientific purpose. The OMIP runs at low and medium resolution are based on the CMCC

climate model system used in the CMIP6 exercises, and closely follow the OMIP experimental protocol (Griffies et al., 2016). The OMIP high-resolution was informally organized by the CLIVAR Ocean Model Development Panel (https://www.clivar.org/clivar-panels/omdp), with no well-defined set-up and spin-up protocols apart from the use of the common forcing. By the time CMCC started the OMIP2 simulations, including GLOB16 code in the framework of the coupled system was not affordable. The differences in the model implementation and set-up impact the results and limit the model intercomparison, but we believe that this model study can still provide insight in the relative benefits and drawbacks of running ocean–sea ice models at eddy-rich resolution, and that the metrics used in the paper are robust enough to highlight the impact of grid refinements, even if not to isolate it.

In this paper, we briefly describe the ocean model and the experiment design (Section. 2). Then, we present GLOB16 climate-relevant ocean variables to provide a general description and evaluation of the global ocean state and the model representation of ocean circulation on global and regional scales (Section 3). First, the temporal evolution of temperature and salinity, upper-ocean temperature, and kinetic energy are presented to examine trends and variabilities in GLOB16, followed by the analyses of the spatial patterns of surface temperature and depth of the mixed layer. Then, ocean surface currents and associated volume transports are provided to highlight the impact of mesoscale dynamics. An overview of sea ice cover is presented for both hemispheres. In Sect. 5, we summarize the study.

## 2 Model and experiment design

GLOB16 is a global, eddying configuration of the ocean and sea ice system built on NEMO modelling framework (https://www.nemo-ocean.eu/). The model is based on its first implementation documented in Iovino et al. (2016), where the ocean component is upgraded from version 3.4 to version 3.6-stable (Madec et al., 2016).

GLOB16 makes use of a nonuniform tripolar grid with a nominal 1/16° horizontal resolution (6.9 km at the equator reducing poleward). The grid consists of an isotropic Mercator grid between 60°S and 20°N, and a non-geographic quasi-isotropic grid north of 20°N. The minimum grid spacing is ~2 km around Victoria Island and the meridional scale factor is fixed at 3 km south of 60°S - the grid has 5762 × 3963 grid points horizontally. Ocean and sea ice are on the same horizontal grid. The vertical coordinate system is based on fixed depth levels and consists of 98 vertical levels with a grid spacing increasing from approximately 1 m near the surface to 160 m in the deep ocean. An outline of the model grid and size is in Table 1 for all models.

The ocean component is a finite difference, hydrostatic, primitive equation ocean general circulation model, with a linearized free sea surface, a free-slip lateral friction condition and Arakawa C grid. Biharmonic viscosity scheme is used in the horizontal directions in the equations of momentum. Lateral tracer diffusion is along isoneutral surfaces using Laplacian mixing. Tracer advection uses a total variance dissipation (TVD) scheme (Zalesak, 1979). Vertical mixing is achieved using the turbulent kinetic energy (TKE) closure scheme (Blanke and Delecluse, 1993). Background coefficients of vertical diffusion and viscosity represent the vertical mixing induced by unresolved processes in the model. Vertical eddy mixing of both momentum and tracers is enhanced in case of static instability. The turbulent closure model does not apply any specific modification in ice-covered regions. Bottom friction is quadratic, and a diffusive bottom boundary layer scheme is included. All configurations use the EOS80 equation of state of seawater (UNESCO, 1983), with potential temperature and practical salinity as prognostic state variables. The ocean component is coupled to the Louvain-la-Neuve sea Ice Model version 2 (LIM2, Timmermann et al., 2005), which has much simpler thermodynamics but also a smaller computational role compared to the more complex LIM3 code (Rousset et al. 2015, Uotila et al. 2017) available in NEMOv3.6. LIM2 is integrated as internal module in the NEMO code, with no need of an external coupling software to process and pass variables between the ocean and sea ice components. Sea ice is solved on the ocean grid. It uses a three-layer model for the vertical heat conduction within snow and ice, features a single sea-ice category and open water represented using ice concentration. The ice dynamics is calculated according to

external forcing from wind stress, ocean stress, and sea surface tilt and internal ice stresses using a C-grid elastic–viscous–plastic rheology (Bouillon et al., 2013).

While the best approach to identify the impact of grid resolution should be to change only resolution and associated physics in the suite of models, this was not the case in similar previous studies (Chassignet et al., 2020, Kiss et al., 2020, Li et al., 2020). We have configured all models independently, following their distinct scientific goals. Given the large computation cost of the GLOB16 configuration, the GLOB16 experiment is configured using our best practices based on the forecasting application (Cipollone et al., 2020, Masina et al., 2021), since it was practically impossible to re-run the code for long sensitivity tests dedicated to the OMIP-2 exercise.

For research and operational applications, CMCC global ocean-sea ice configurations at low- and medium-resolution generally follow the GLOB16 framework, with the ocean component coupled to the sea ice module integrated in the NEMO system. Here, for the OMIP exercise, the non-eddying (ORCA1) and eddy-permitting (ORCA025) ocean models are derived from the long CMCC experience in coupled climate modeling. The two configurations constitute the ocean-sea ice component of the coupled CMCC Climate Model (CMCC-CM2, Cherchi et al., 2019) and Earth System Model (CMCC-ESM2, Lovato et al., 2022). This model system is based on the Community Earth System Model (CESMv1.2), in which we replaced the original ocean component by NEMOv3.6 (Fogli and Iovino 2014). The ocean component is coupled to the Community Ice Code CICEv4.1 (Hunke and Lipscomb, 2010) via the cpl7 coupling architecture. ORCA1 has a 1° tripolar horizontal mesh with additional meridional refinement up to 1/3° in the equatorial region, while ORCA025 has a nominal resolution of 1/4°, both with 50 vertical levels, ranging from 1 to 400 m. The ORCA1 physical core as implemented for the OMIP2 simulation is described in Tsujino et al. 2020. It is shared with ORCA025 code except for resolution-dependent features, such as the eddy induced tracer advection term (Gent and McWilliams, 1990) added in ORCA1, not in ORCA025. ORCA1 also employs a strong no-slip condition to reduce the transports through narrow straits. The sea ice model includes energy-conserving thermodynamics (Bitz and Lipscomb, 1999), multi-category ice thickness (Bitz et al., 2001) with 5 thickness categories, and elastic-viscous-plastic ice dynamics (Hunke and Dukowicz, 1997). The sea ice model is solved on the Arakawa B-grid, with the tracer points aligned with the ocean grid. The coupling interface between NEMO and CICE is described in Cherchi et al. (2019) and references therein (Fogli and Iovino, 2014). To be able to attribute the main differences among model configurations mainly to the increase of ocean resolution in the horizontal and vertical grids, the three configurations employ, as far as possible, the same numerical schemes and parameterizations, except grid-spacing dependent parameters. Key changes in the ocean parameters setting are listed in Table 1.

**Table 1. Outline of grid characteristics, ocean/ice timesteps and physical parameters used for the model simulations, together with the computational performance (the number of cores used by the ice models are indicated in parenthesis).**

| Parameter | Low resolution ORCA1 | | Medium resolution ORCA025 | | High resolution GLOB16 | |
|---|---|---|---|---|---|---|
| Horizontal grid points | 360×291 | | 1440×1050 | | 5760×3962 | |
| Lateral spacing | 1° | | 0.25° | | 0.0625° | |
| number of vertical levels (n) | 50 | | 50 | | 98 | |
| maximum depth \| depth at level n/2 [m] | 5904 | 252 | 5904 | 252 | 6181 | 504 |
| Surface \| bottom level spacing [m] | 1.05 | 410 | 1.05 | 410 | 0.8 | 162 |
| Ocean baroclinic time step [sec] | 3600 | | 1200 | | 200 | |
| Ocean-ice coupling timestep [sec] | 3600 | | 1200 | | 600 | |
| Ice model dynamic - thermodynamic time steps [sec] | 30 - 3600 | | 10 - 1200 | | 5 - 600 | |
| Horizontal viscosity | Laplacian $10^4$ m$^2$ s$^{-1}$ | | Biharmonic -1.8 $10^{11}$ m$^4$ s$^{-1}$ | | Biharmonic -0.5 $10^{10}$ m$^4$ s$^{-1}$ | |

| | | | |
|---|---|---|---|
| Tracer diffusivity [m$^2$ s$^{-1}$] | $10^3$ | 300 | 80 |
| Vertical viscosity [m$^2$ s$^{-1}$] | $10^{-4}$ | 1.2 $10^{-4}$ | 1.2 $10^{-4}$ |
| Vertical diffusivity [m$^2$ s$^{-1}$] | $10^{-5}$ | 1.2 $10^{-5}$ | 1.2 $10^{-5}$ |
| Eddy parameterization | Yes | no | no |
| Eddy induced velocity coeff. [m$^2$ s$^{-1}$] | $10^3$ | - | - |
| Number of cores for the ocean (sea ice) component | 128 (96) | 1008 (972) | 2086 |
| Wall time (h yr$^{-1}$) | 1.31 | 4.44 | 94.22 |

All three simulations are forced by the version 1.4 of the JRA55-do dataset whose temporal coverage extends from January 1958 to near present. We use 1958-2018 for all runs used in this manuscript. JRA55 temporal and horizontal resolutions are 3-hours and 0.5625° (55km), respectively. The dataset includes liquid and solid precipitation, downward surface longwave and shortwave radiation, sea level pressure, 10 m wind velocity components, 10 m specific humidity, and 10 m air temperature.

The Large and Yeager (2004) turbulent flux bulk formulas are used in all three configurations to calculate turbulence heat and momentum fluxes. Wind velocity in JRA55-do has been adjusted to match time-mean scatterometer and radiometer winds, which are relative to the ocean surface current. Tsujino et al. (2018) recommended to add a climatological mean surface current to JRA55-do winds to better represent absolute winds. However, since this approach was not tested yet, we did not apply it. We use the wind velocity relative to the full ocean surface velocity in the calculation of wind stress (relative wind stress) on

the ocean and on sea ice. GLOB16 uses a bilinear interpolation for all variables but the wind which uses a bicubic interpolation. ORCA1 and ORCA025 use the default CESM interpolation methods based on Earth System Modeling Framework (ESMF, https://www.earthsystemmodeling.org/). In particular, state variables (temperature, pressure, etc...) are interpolated using a bilinear interpolation, fluxes using a first-order conservative remapping and vectors using a higher-order patch recovery (a second-degree polynomial re-gridding method, which uses a least squares algorithm to calculate the polynomial).

JRA55-do provides also the total freshwater discharge at 0.25° resolution, it consists of the daily and interannually varying continental river runoff (Suzuki et al., 2018), the monthly freshwater from ice sheets and glaciers in Greenland (Bamber et al., 2018) and the climatological estimates of Antarctic calving and basal melt (Depoorter et al., 2013). Liquid runoff is deposited along the coast and distributed in the upper 20 m in the lower-resolution runs, at the ocean surface in GLOB16, with no specific enhancement of the mixing in all cases. The runoff interpolation in all three configurations makes use of a globally conserving

method which also spreads the runoff along the coast, to compute offline remapping weights.

The experiments ran for different time lengths. While the 1° and 1/4° experiments were performed for six 61-year cycles (1 January 1958–31 December 2018) of JRA55-do, GLOB16 was integrated over a single cycle. The GLOB16 grid has much higher resolution than the forcing data that implies that mesoscale activities are produced from the internally generated variability. Only the first JRA55-do cycle is analyzed for all simulations in this paper.

As suggested by the OMIP-2 protocol, the ocean was initially at rest, with zero sea level and with temperature and salinity from the World Ocean Atlas 2013 v2 (WOA13, Locarnini et al., 2013, Zweng et al., 2013) "decav" product (averaged from 1955-2012) interpolated on a 0.25° grid. The initial sea ice conditions are different among models: the initial sea ice properties in ORCA1 and ORCA025 runs are taken from spin-up experiments, while ice concentration and thickness for GLOB16 are fixed to 100 % and about 3 m (1 m), respectively in regions north of 70° N and south of 60°S.

We restore sea surface salinity (SSS) to the WOA13 v2 monthly climatology. Salinity restoring is applied globally via an equivalent surface freshwater flux. There is no salinity restoring under sea ice covered areas. The timescale is set by the "piston velocity" (surface vertical grid spacing divided by restoring timescale) of one year over the upper layer of nominal 100m thickness (100m/1y) in ORCA1 and ORCA025 cases, over 50 m thickness (50m/1y) in GLOB16. It is important to mention that the two sea ice models used in our two systems employ different bulk salinity affecting the salt release from the sea ice to

the ocean. In CICE, a reference value of ice salinity (4 psu) is used for computing the ice-ocean exchanges, although the ice salinity used in the thermodynamic calculation has different values in the ice layers with the vertical salinity profile prescribed and fixed in time. In our version of LIM2, the fresh water (salinity) fluxes between the ice and the ocean assume constant salinities of 6 psu. Over a sea ice formation and melt cycle, this produces stratification differences among runs and might have an impact on the large-scale ocean circulation.

## 2.1 Computational performance

All simulations were performed on the CMCC Zeus HPC platform, equipped with two Intel Xeon Gold 6154 (3.0 GHz, 18 cores) and 96 GB of main memory per node. The interconnection network is the 100Gbps Infiniband EDR, while the file system is the IBM General Parallel File System (GPFS). The models were compiled with the Intel compiler suite version 20.1 and MPI library (based on MPI version 3.1). The computational performance of the three configurations during the OMIP2 production runs are in Table 1. Given the limited computational resources available and the need to divide these resources among the three simulations, these are not fully representative of the best achievable performances. It is worth mentioning that the computational performance of a coupled modeling system as the one used for ORCA1 and ORCA025 depends on the performances of any single model component and the efficiency of the coupling software. The NEMO and CICE codes run sequentially. In the lowest-resolution, NEMO uses ~78% and CICE ~7.5% of the wall time; the remaining is given to the atmospheric and river data models and to the coupler. In ORCA025, the ocean and ice models use 72% and 15.3%, respectively. In the GLOB16 framework, LIM is not a stand-alone model, it is a module of the ocean code. The two components are interactively interfaced without using a coupling code; LIM2 takes almost 20% of the wall time. The computational performances of the GLOB16 configuration are affected by the large amount of data that needs to be transferred at each model time step between the main memory and the CPUs, together with the associated burden on the cache memories hierarchy, and by the inter-process communication through the MPI library. Finally, these can be further reduced by the input/output operations, especially when performed at daily frequency.

## 3 Model evaluation

### 3.1 Temporal evolution

In this section, we characterize features of the temperature and salinity drift as developed in the GLOB16 run in comparison to observational datasets and the lower resolution configurations. The time evolution of the volume-weighted annual mean global ocean potential temperature is shown in Figure 1a. All timeseries start with similar temperature close to the WOA13 initial state (~3.59°C). Then, there is a heat uptake in GLOB16 with the ocean warming up from the beginning of the run, achieving a warming of 0.05°C at the end of the integration. In the medium- and low-resolution oceans, the volume mean temperature gradually decreases in the first ~40 years to warm up thereafter, but staying cooler than the initial condition. Similar behavior is reproduced in the following cycles of ORCA1 and ORCA025, with an overall cooling of ~0.15°C over six cycles (not shown), in agreement with the cooling of the low-resolution OMIP2 ensemble mean (Tsujino et al., 2020). The effect of resolution on the thermal evolution of the entire water column is the consequence of different model responses at different depths in the ocean interior (Fig. 2). Comparing the same model at different horizontal resolution and different models at similar resolution does not underline a coherent behavior for the temperature evolution in OMIP2-like simulations. In Tsujino et al. (2020), the spread among non-eddying models is 0.3°C wide with five (four) models out of eleven showing increasing (decreasing) temperatures, two with no drift after one cycle. A comparable spread is found for OMIP1 and CORE-II runs (Tsujino et al., 2020, Griffies et al., 2014). In Chassignet et al. (2020), all eddy-rich models present an increase in global temperature, the spread among the four eddy-rich models is ~0.1°C, and the increase in horizontal resolution does not necessarily result in a reduction in temperature drift.

The temporal evolution of the annual globally averaged sea surface temperature (SST) is well reproduced by the three models (Fig. 1b), being largely affected by the shared atmospheric properties. The simulated SST records present similar interannual variability and lie between the observation-based estimates. Three validated datasets have been used: the global 2°×2° Extended Reconstructed Sea Surface Temperature (ERSSTv5, Huang et al., 2017), the global 1°×1° Hadley Centre Global Sea Ice and Sea Surface Temperature (HadISSTv1.1, Rayner et al., 2003) and the SST from the European Space Agency (ESA)

Climate Change Initiative (CCI) Programme derived from satellite observations dataset and gridded on a 0.05° × 0.05° global mesh from 1981 upward (Atkinson et al., 2014; Good et al., 2019). GLOB16 is slightly warmer than the other two models (note that ORCA1 and ORCA025 SST coincide in Fig. 1b), and the temperature increase over the integration period is roughly 0.5°C. The slightly smaller variation of the lower-resolution SST agrees with results in Tsujino et al. (2020), with an increase of 0.4°C. The impact of resolution on the simulated SST does not correspond to what was found in the OMIP2 comparison by

Li et al. (2020) and Kiss et al. (2020) where the 1/10° ocean surface is the coldest. All three models capture the expected warming trend that is higher and closer to the observations in GLOB16 with a value of about 0.074°C/decade during 1958–2018, and 0.084°C/decade during 1982–2018. The GLOB16 trends are slightly smaller than those from ErSST and ESACCI SST and very close to HadISST.

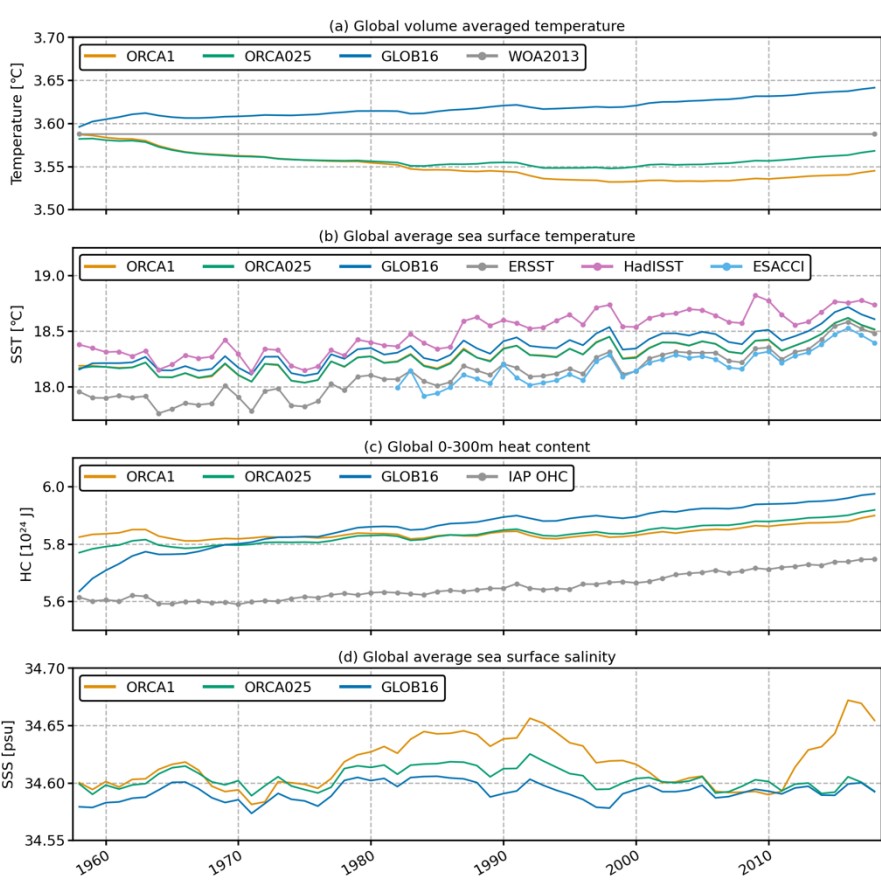

**Figure 1.** Time evolution of the global annual-mean (a) volume-weighted ocean temperature (°C), (b) sea surface temperature (°C ) compared with observed HadISST in violet, ErSST in gray, ESA CCI SST in cyan, (c) ocean heat content (J) integrated in the depth range 0-300m and (d) sea surface salinity (psu) for GLOB16 and the lower-resolution models, during the first integration cycle from 1958 to 2018.

The GLOB16 heat uptake is better explained in Figure 2 that shows the evolution of the annual mean anomaly of the global horizontally-averaged temperature as a function of depth. This metric shows to what extent and how quickly the modelled 3D temperature deviates from the ocean initial state as the resolution changes. The anomaly for a specific date is computed as the difference between this current value and the WOA13 temperature. While the vertical structure is not greatly affected by the

255 resolution, there are large changes in the magnitude of differences among configurations. There is a strong depth-dependent

thermal adjustment from the initial condition in GLOB16 (Fig. 2c) that exhibits a large and rapid subsurface warming down to 500m from the beginning of the simulation. This warming is centered within the 100–200m depth range and the maximum error is larger than 1°C. GLOB16 shows a weak and gradual cooling in the mid-depth and deep ocean. This upper ocean warming is found in other eddying oceans (e.g. Lellouche et al., 2021; Chassignet et al., 2020), but the impact of resolution on the temperature drift is largely model dependent (Kiss et al., 2020, Chassignet et al., 2020) and might be due to different resolved and parameterized processes.

Our lower resolution configurations show smaller changes from the initial condition as a function of depth; the warming in the upper hundred meters is less pronounced and slower. The analysis in Tsujino et al. (2020) shows that the temperature drift in the low-resolution simulations continuously deepens and strengthens at all depths during the six forcing cycles.

The thermal adjustment in GLOB16 is consistent with the significant increase in the SST (Fig. 1b) and ocean heat content (OHC) integrated in the 0-300m depth range (Fig. 1c). The OHC, as an indicator for this heat accumulation (e.g., Cheng et al., 2021), is slightly larger than estimates from the Institute of Atmospheric Physics (IAP, Cheng et al., 2017), and its temporal evolution is approximately linear in all runs. The mean OHC in GLOB16 has a linear warming of $3.096 \times 10^{22}$ J per decade that closely follows the observed one ($3.033 \times 10^{22}$ J/decade) from 1970 (after the large models' adjustment) to 2018 (Table 2).

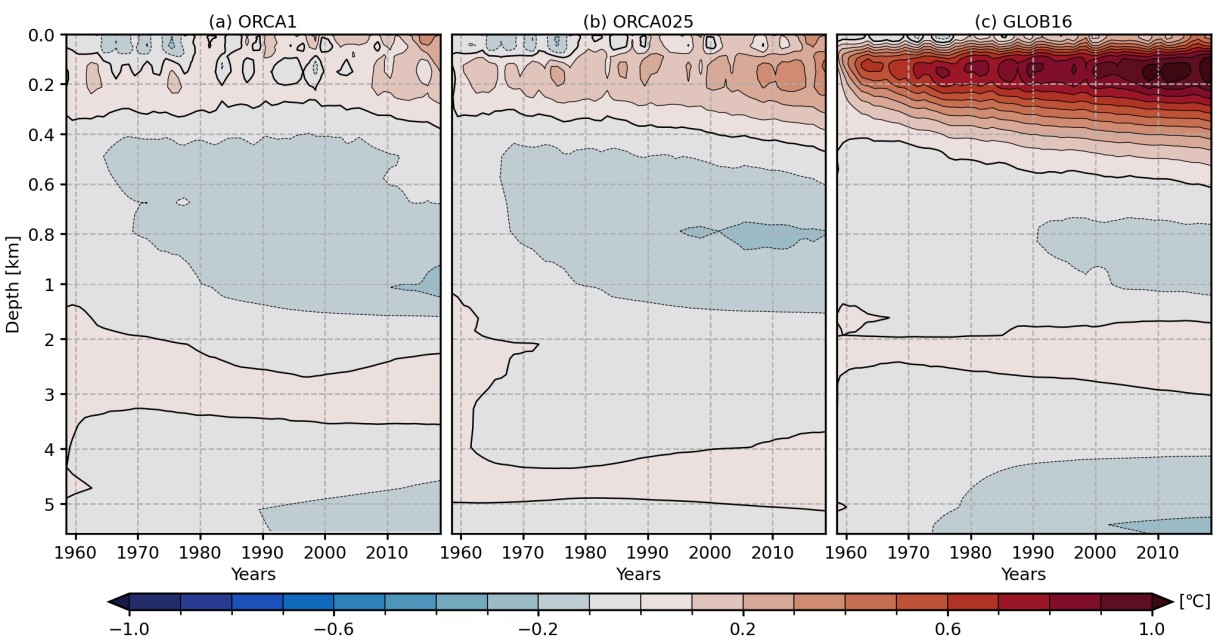

**Figure 2. Time evolution of the annual-mean anomalies (relative to WOA13) of the horizontally averaged potential temperature (in °C) as a function of depth and time from 1958 to 2018. The upper 1000m are stretched, and 0.1°C contours for ΔT are drawn.**

Figure 1d displays the time series of annual mean sea surface salinity, averaged over the global domain. All models remain relatively stable with similar interannual variability except ORCA1 that is generally the most saline ocean and overestimates the other runs in 3 decades. The SSS drift is offset by the surface salinity restoring that is incorporated into the codes to constrain the salinity drift in the model ocean (in Sect. 2). The restoring of SSS drives its quasi-stationary evolution, and the salt exchange between ocean and sea-ice due to ice formation and melting, is the only source of salt for the ocean. Compared to recent Argo salinity observations that have a mean value of ~34.9 psu, all simulations present fresher surface ocean as generally seen in OMIP2 runs (Tsujino et al. 2020, Kiss et al. 2020, Li et al. 2020), suggesting differences between the observational datasets and the WOA initial conditions.

**Table 2. Global annual mean, its standard deviation and linear trend of sea surface temperature for the period 1982-2018 (common to all SST datasets), and of the 0-300m OHC for the period 1970-2018.**

| SST | ORCA1 | ORCA025 | GLOB16 | HadISST | ErSST | ESACCI SST |
|---|---|---|---|---|---|---|
| Annual mean (℃) | 18.351 | 18.349 | 18.432 | 18.604 | 18.238 | 18.187 |
| Standard deviation (℃) | 0.10 | 0.11 | 0.11 | 0.12 | 0.14 | 0.15 |
| Linear Trend (°C/dec) | 0.076 | 0.079 | 0.084 | 0.080 | 0.116 | 0.121 |

| OHC | ORCA1 | ORCA025 | GLOB16 | IAP | | |
|---|---|---|---|---|---|---|
| Annual mean ($10^{24}$ J) | 5.840 | 5.845 | 5.889 | 5.662 | | |
| Standard deviation ($10^{22}$ J) | 2.05 | 3.1 | 3.31 | 4.43 | | |
| Linear Trend ($10^{22}$ J/dec) | 1.175 | 2.058 | 3.096 | 3.033 | | |

Figure 3 shows the time evolution of the global-averaged kinetic energy (KE) for the three configurations from 1958 to 2018. The KE evolution is similar between models with a quick increase in the first two years, to slowly decrease in the following decades and level off at the end of the integration period. The global KE is a strong function of resolution and is expected to be higher in oceans that contain more turbulent processes; the total KE from the low to high resolution model increases by a factor of ~4 (as in Chassignet et al., 2020) and the eddying ocean still underestimates by more than half the observation-based estimates (Chassignet and Xu, 2017). While ORCA1 quickly reaches a steady state of ~ 4.5 cm$^2$ s$^{-2}$, KE in ORCA025 and GLOB16, with a mean value of ~14.2 cm$^2$ s$^{-2}$ and ~18.5 cm$^2$ s$^{-2}$ respectively, decreases of ~10% by 2018. The global averaged eddy component of KE (defined as the kinetic energy of the time-varying component of the velocity field) contributes to the total KE by about 65% in GLOB16, 50% in ORCA025 and only 25% in ORCA1 (not shown).

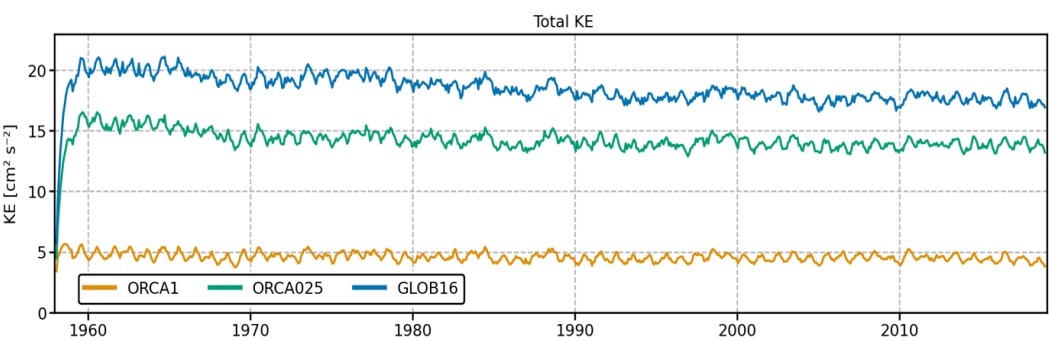

**Figure 3. Time series of the global average of monthly-mean total kinetic energy (in cm$^2$ sec$^{-2}$) during the whole integration cycle, from 1958 to 2018.**

## 3.2 Horizontal spatial distribution

### 3.2.1 Sea surface temperature and sea surface height

To further examine the surface temperature differences, Figure 4 shows latitude-longitude maps of SST biases computed with respect to ERSSTv5, over the last 10 years of the integration (2009-2018). Overall, the large-scale pattern of the thermal error is similar among configurations, suggesting possible systematic biases in the initial or surface boundary conditions or surface forcing. The largest SST differences from observations are collocated with energetic eddy activity and major frontal zones, where SST gradients are strongest, and the shift of jet locations results in large biases. GLOB16 still presents some of the common model biases, but most of the SST biases are reduced when horizontal resolution increases. While the globally averaged SST error is similar among models (0.52℃ for ORCA1, 0.48℃ for ORCA025 and 0.50℃ for GLOB16), there are clear improvements at local scales. For instance, the warm biases associated with western boundary currents (WBCs) seen in most of the OMIP2 models at non-eddying resolution (Tsujino et al., 2020; Chassignet et al., 2020) are significantly reduced.

Clear improvements are also seen in the North Atlantic where the cold bias in the southern subpolar gyre weakens and covers a much smaller area, due to a more realistic representation of the North Atlantic Current and convection processes (Danabasoglu et al., 2014). On the other hand, the cold bias in the Nordic Sea is stronger in GLOB16 presumably due to changes in the northward transports. Generally, GLOB16 has warmer SST in tropical and subtropical regions compared to ORCA1 and ORCA025. The SST does not benefit from the resolution in the eastern boundary upwelling regions where the bias has been shown to be sensitive to atmospheric forcing resolution (Tsujino et al., 2020, Bonino et al., 2019). In the Southern Ocean, biases are larger in the energetic regions and are reduced in GLOB16 compared to the lower-resolution experiments due to a more realistic representation of fronts.

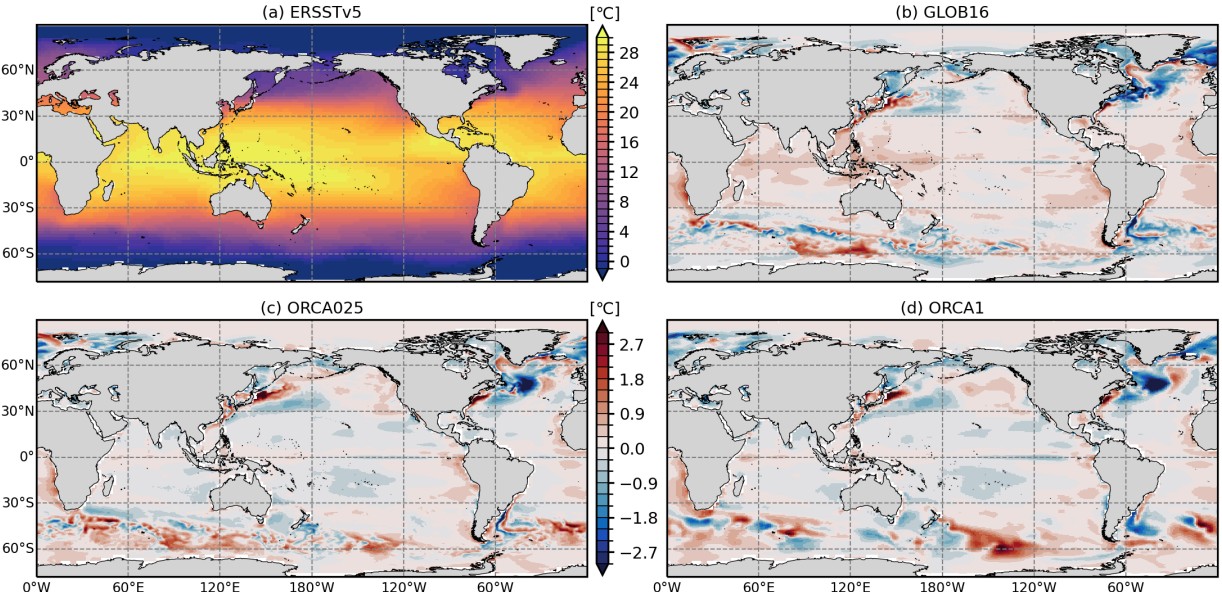

**Figure 4. Model sea surface temperature differences (in °C) from ERSSTv5 averaged over the 2009-2018 period.**

To assess the dynamical capacity of the model configurations and to evaluate the benefit of increased grid resolution in representing surface mesoscale activity, Figure 5 presents the spatial patterns of the sea surface height (SSH) variability, represented by the plots of the standard deviation (STD) of daily SSH anomaly, for the multi-mission satellite derived sea level product (AVISO SSALTO/DUACS, https://www.aviso.altimetry.fr/) and each model configuration, for the period 2009-2018. High SSH variability is typically located in regions populated by the energetic mesoscale eddies. In AVISO (Fig. 5a), large variability is collocated with the WBC systems and their jet extensions, the strong equatorial current system, as well as the Brazil and Malvinas current system, the Agulhas Current, and the Antarctic Circumpolar Current (ACC) in the Southern Ocean. This variability is associated with high kinetic energy. The eddy-rich GLOB16 shows a significant improvement in the position, strength, and variability of the western boundary currents, the ACC and the Zapiola gyre. The magnitude of SSH variability is the closest to what is estimated from altimetry. ORCA025 is also able to capture the general observed variability but presents weaker flow instabilities and fewer meanders. Since eddies are not explicitly resolved in the ORCA1 configuration, it does not represent any significant SSH variability that decreases substantially in all the global domain. It is worth noting that all models reproduce the spatial pattern but underestimate the amplitude of the SSH variability in the tropical Pacific-Indian Ocean (associated with El Niño-Southern Oscillation).

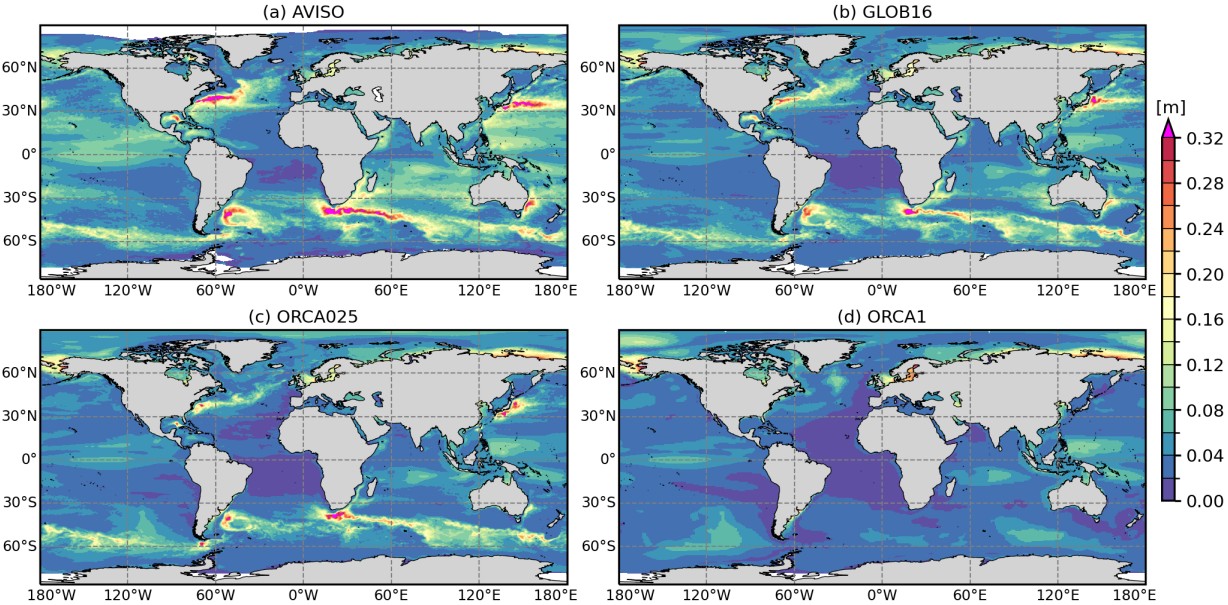

**Figure 5. Standard deviation of sea level anomaly for the year 2009-2018 for the three simulations and the AVISO SSALTO/DUACS gridded analysis of satellite altimetry.**

### 3.2.2 Mixed layer depth

Here, we analyze the GLOB16 simulated mixed layer depth (MLD) in the boreal and austral winters (Figure 6), when the mixed layer reflects the depth of rapid overturn of surface water which is closely related to formation of dense and deep-water masses. The MLD is shown for the March and September climatologies in the Northern hemisphere (NH) and Southern hemisphere (SH), respectively, computed over the last 10 years of the model integration, and validated against observed estimates (the lower-resolution models are also shown for comparison). The model values are validated against a recent dataset of monthly climatology of surface MLD over the global ocean, which is computed from 4.5 million hydrographic profiles from the NCEI-NOAA World Ocean Database (WOD) and the Argo program (de Boyer Montégut, 2022). The observed MLD is diagnosed through a density threshold criterion as the depth over which the potential density increases by 0.03 kg m$^{-3}$ from the reference value of surface potential density taken at 10m depth; resulting values are mapped on a monthly basis at 1°x1° spatial resolution (de Boyer Montégut et al., 2004). The same density threshold method is applied to model output.

The GLOB16 winter MLD is highly variable in space and time and presents a strong seasonal cycle, as in observed fields (de Boyer Montegut et al., 2022; Johnson and Lyman, 2022; Holte et al., 2017). In March, the NH GLOB16 shows a spatial pattern close to the observed one, with good correspondence between regions of shallow and deep mixed layers. Both the modelled and observed fields show regions of shallow MLD at low latitudes. The mixed layer deepening at mid-high latitudes is highly heterogeneous in space and well reproduced in both hemispheres. In general, in regions of strong convection (e.g., North Atlantic subpolar regions, Weddell and Ross and south-eastern Pacific) the GLOB16 mixed layer is deeper than observation-based estimates. This mismatch between the model and observations can depend on limitations of the model physics, but it is also worth noting that the comparison with the observed dataset is less robust at high latitudes due to the scarcity of in situ ocean observations in winter. Deeper mixed layer is simulated along the ACC where its depth can reach ~150m. In the North Atlantic subpolar gyre, GLOB16 reasonably simulates, in terms of depth and location, the winter deep mixed layer associated with the North Atlantic Deep Water (NADW) formation, thus it has the capability to form water masses at the right locations. In the subpolar gyre and Nordic Seas, the GLOB16 penetration depth compares well with the observations, with the closest agreement in the Irminger Sea and a negative bias in the Greenland Sea. As seen in other high-resolution models (Treguier et al., 2023), the high-resolution model overestimates the observed 600m mixed layer within the Labrador Sea basin, where it exceeds 1000m. There is a strong dependence of the MLD on the spatial grid resolution in the northern high-latitude ocean

sectors. The ORCA1 model tends to overestimate the amplitude and the location of MLD maxima in the Nordic Seas and Irminger Sea with weak convection in the Labrador Sea (Fig. 6d), as generally happens in non-eddying oceans (e.g. Tsujino et al., 2020, Brodeau and Koenigk, 2016, Danabasoglu et al., 2014). Increasing to eddy-permitting resolution in ORCA025, the MLD is reduced north of the Greenland Scotland Ridge, but largely deepens in the Labrador Sea with also a too wide horizontal extension of convection (Fig. 6c), compared to observations (e.g. Koenigk et al., 2021). These well-known features

in lower-resolution ocean models appear largely improved in eddying oceans able to resolve the key mesoscale processes that strongly control the stratification, and intensity of the Labrador Sea Water production (Pennelly and Myers, 2020). GLOB16 simulates deep mixed layers in the sea-ice cover areas in the coastal Weddell Sea and Ross Sea, yielding persistent winter convective overturning off Antarctica with implications on the rate of the Antarctic Bottom Water (AABW) formation. Only few observed profiles are present there, but the winter mixed layer at the shelf break is found locally to be 300–500 m deep

from the under-ice Argo network (Pellichero et al., 2016). In ORCA1 and ORCA025, the mixed layer deepens in the Weddell Sea gyre and all along the Antarctica coastlines. It is well known that although the AABW is formed over the continental shelves and then sinks to the bottom along the Antarctic slope, low-resolution ocean models generally reproduce unrealistic deep mixed layers in the Weddell Sea where the AABW is formed by open-ocean convection in the gyre (e.g. Heuzé 2021). To a great extent, these differences in the mixed layer structure around Antarctica may have a dynamical origin, but they might

also be due to the sea ice formation and brine rejection that follow different schemes in GLOB16 (with LIM2 sea ice model), and ORCA runs (with CICE sea ice model).

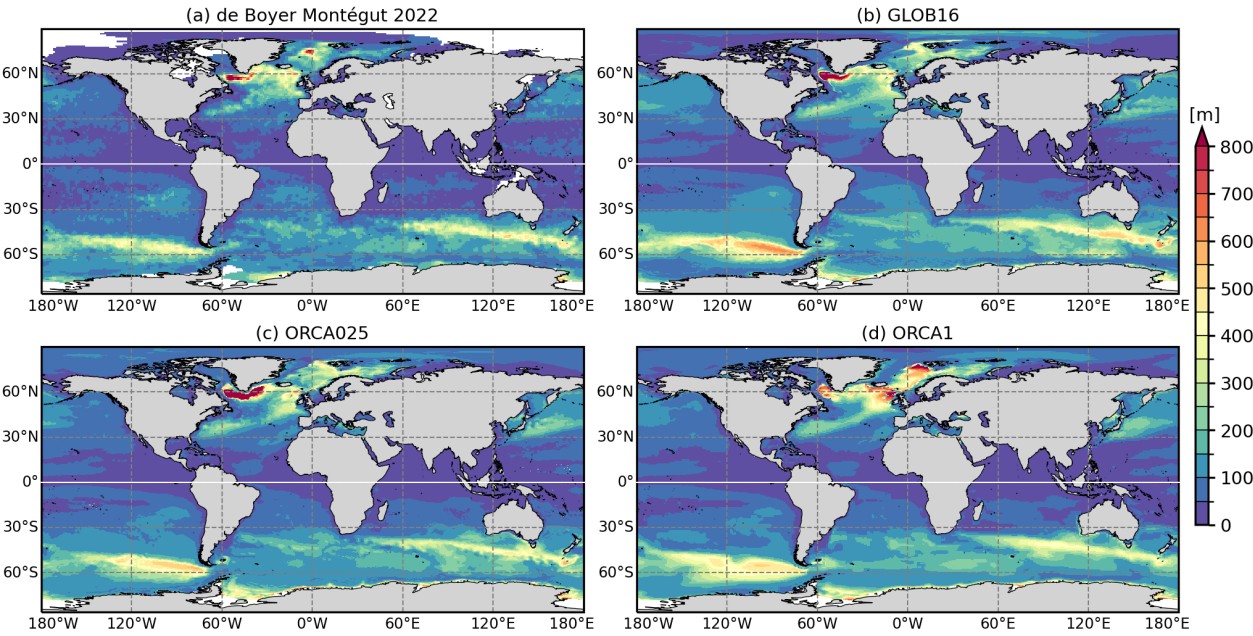

**Figure 6. Mean mixed layer depth (in m) averaged over March (in the Northern Hemisphere) and September (in the Southern**
**Hemisphere) from (a) the de Boyer Montégut et al. (2022) climatology, (b) GLOB16, (c) ORCA025 and (d) ORCA1. All MLD fields are computed as the monthly climatology over last 10-year output.**

In September, the observed shallow ML in the NH is well reproduced in all models (Figure 7), with a marked sign of the upper-ocean circulation on the modelled spatial distribution. In the Southern Ocean, observations show mixed layer deepening
north of the marginal sea-ice zone, toward the ACC (Fig. 7a) to reach the very deep convection area associated with the formation of mode water. GLOB16 reproduces this band of deep mixed layers extending along the northern ACC flank in the Indian and Pacific Oceans (Fig. 7b), but generally overestimates the observed penetration depth and lower-resolution models (Fig 7c, d) in the Eastern Pacific.

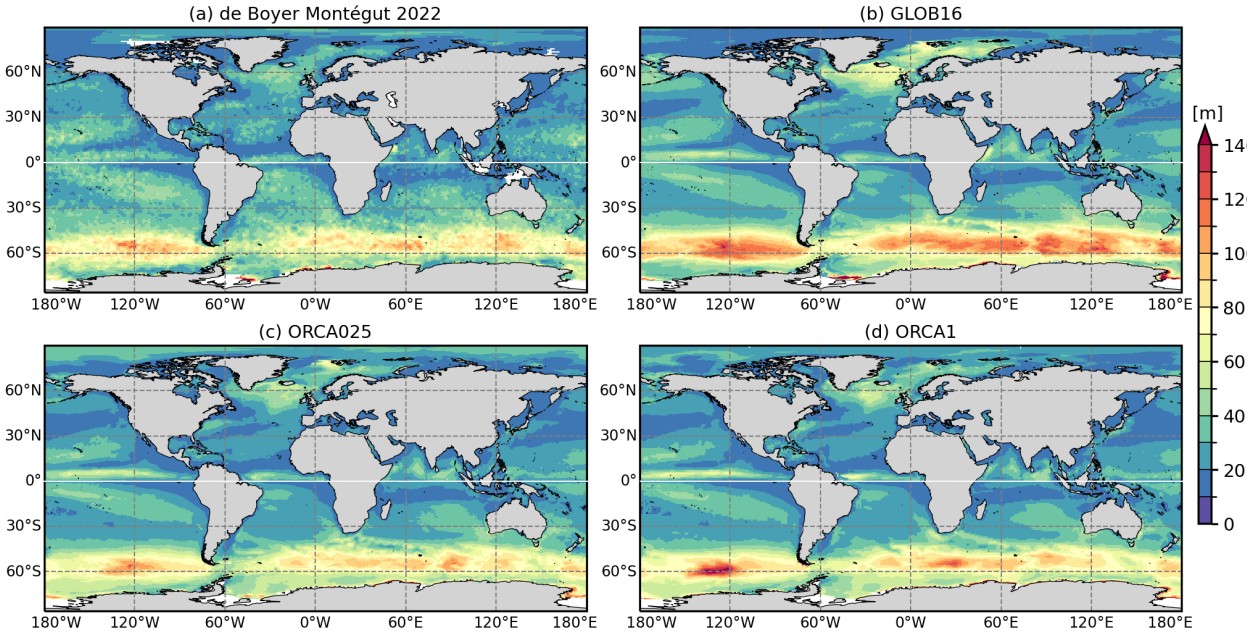

**Figure 7. Mean mixed layer depth (in m) averaged over September (in the Northern Hemisphere) and March (in the Southern Hemisphere) from (a) the de Boyer Montégut et al. (2022) climatology , (b) GLOB16, (c) ORCA025 and (d) ORCA1. MLD fields are computed as the monthly climatology over last 10-year output.**

To better illustrate differences in the modelled and observed MLD, Figure 8 presents the zonal mean of the March and September MLD climatology as a function of latitudes (between 70°S and 85°N). A second data set is also used for an overview of the zonal mean MLD biases. Johnson and Lyman (2022) have recently published a statistical monthly climatology of the Global Ocean Surface Mixed Layer (GOSML, https://www.pmel.noaa.gov/gosml/) based on ARGO data. They find that the distribution of MLD is non gaussian, with large skewness and kurtosis that vary seasonally and spatially. The MLD variance also displays seasonal variations, and it depends on the MLD itself (regions with large MLDs have a large MLD variance).

As global zonal mean, GLOB16 has a generally larger or similar mixed layer depth compared to lower resolution models. As expected, the March MLD differences between GLOB16 and the other two models are much larger in the northern hemisphere where GLOB16 ML starts to deepen at ~15°N with a clear increase at the WBC latitudes. In the subpolar gyre the MLD differences between high- and low-resolution models are as large as the differences between the two observation datasets. GLOB16 is in close agreement with GOSML estimates from 55° to 65°N northward, with mixed layer by de Boyer Montegut (2022) the shallowest and ORCA025 that overestimates all products and models. Maximum North Atlantic MLD is clearly mislocated in ORCA1. In September, all models overestimate the observed MLD in the Southern Ocean up to ~25°S. In the rest of the basin, all models follow the shallow mixed layer and again the model spread is comparable to the spread of observations. GLOB16 is the closest to GOSML from 40°N northward.

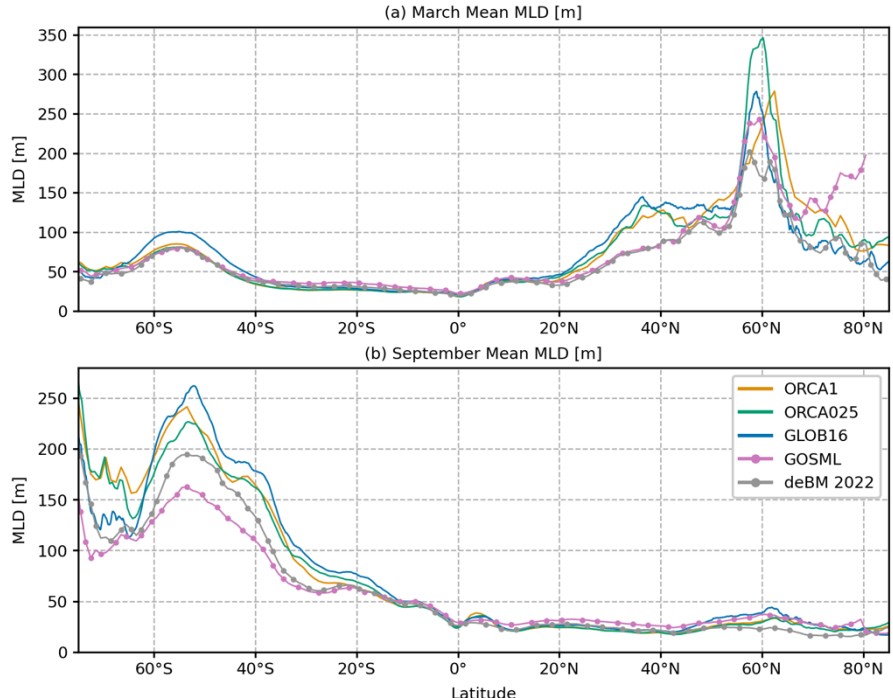

**Figure 8. Zonal mean MLD (in m) as a function of latitude between 75°S and 85°N in the three models and two observation-based estimates, GOSML (Johnson and Lyman, 2022) in pink and de Boyer Montegut (2022) in grey, for a) March and b) September averaged from 2009-2018.**

## 3.3 Ocean circulation

### 3.3.1 Near-surface ocean currents

We compare maps of the GLOB16 ocean current with the Ocean Surface Current Analyses Real-time (OSCAR; http://podaac.jpl.nasa.gov) dataset for the full modeled domain (Fig. 9), and zoomed into the key dynamical regions (Figs. 10 and 11). The OSCAR field is calculated from satellite datasets and consists of a geostrophic term, a wind-driven term, and a thermal wind adjustment, vertically averaged over a surface layer thickness of 30 m and interpolated on a 0.25° grid. The lower resolution models are also shown. Comparison is made over the last 10 years of the cycle integration using daily output. Globally, the large-scale current system represented by GLOB16 qualitatively compares very well with observations, with the model reproducing each of the local maxima in OSCAR. The large dynamic systems and their amplitude are sharply reproduced: the WBCs (such as the Kuroshio, Gulf Stream, North Brazil Current), the Loop Current in the Gulf of Mexico, the Agulhas recirculation, the Leeuwin Current, the Zapiola anticyclone and Antarctic Circumpolar Current (ACC). Despite the improvement in regions of strong and unstable currents, GLOB16 underestimates observed estimates in specific regions as in the equatorial current system (10°S-10°N), but its current velocity is higher than lower resolution runs over the entire domain. Although the eddy-permitting model reproduces the spatial pattern of satellite estimates (Fig. 9c), the intensity of the global current system is overall lower than GLOB16 and OSCAR. The spatial distribution of the upper ocean current system represents regions of intense activity concentrated along well-known ocean surface currents, including WBCs (with a limited extension), and bands of strong activity represented in the tropics and ACC region. Nonetheless, there are regions in which the ocean current are underrepresented, such as the East Australian Current and the Mozambique Channel. As many of the coarse ocean components of CMIP5/6 models, the ORCA1 configuration shows a clearly poorer representation of the surface current systems at global scale (Fig. 9d), compared to the eddy-rich and eddy-permitting models. It captures the major current systems of the global ocean, but it underestimates the magnitude of the surface velocity and fails to represent mesoscale eddies and meanders.

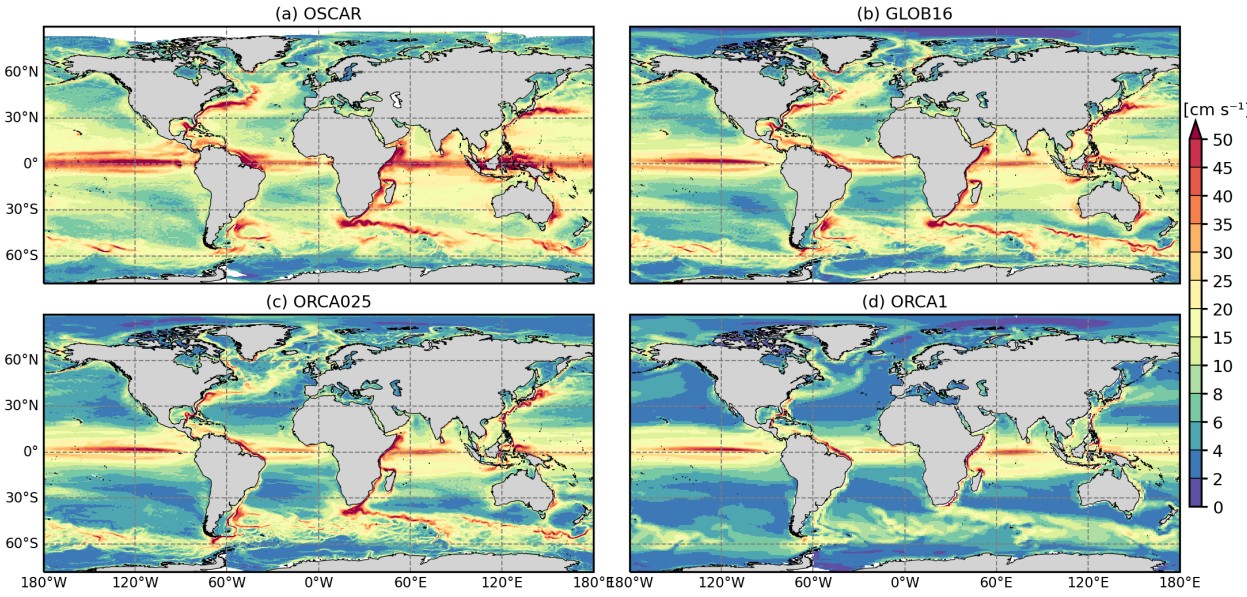

**Figure 9. Ocean current (in cm s$^{-1}$) averaged between 0-30m of the global domain for the three simulations and the OSCARv3 data set.**

It is widely recognized that the horizontal grid spacing, sufficient to resolve the Rossby radius of deformation in most of the global domain and allow for a proper representation of baroclinic instability, results in a significant improvement in western boundary currents and associated eddies (e.g. Hurlburt and Hogan, 2000; Yu et al., 2012; Chassignet and Xu, 2017). A proper representation of the WBCs in global ocean models is the result of many contributing factors. Despite the general improvements in their representation due to model resolution, the simulated WBCs strength, width, position and separation remain dependent on a variety of parameter choices made in the numerical models (e.g. Bryan et al. 2007; Chassignet and Marshall 2008), such as boundary conditions, coastline and bottom geometry, friction parametrization, etc. Accurately simulating the Gulf Stream separation in ocean numerical models has been a challenge and does still remain an issue despite the fact that major improvements are realized in eddy-rich ocean configurations (e.g., Chassignet and Xu, 2017). The Gulf Stream simulated by the three models is presented in Figure 10 (left column). Observations show that the Gulf Stream separates from the coast at Cape Hatteras (35°N, 75°W), and the North Atlantic Current flows north along the east side of the Grand Banks from 40° to 50°N (e.g. Rossby, 1996). Consistent with other modeling studies (e.g., Chassignet and Xu, 2021, Kiss et al., 2020, Petersen et al., 2019), the eddy rich GLOB16 presents a considerable improvement in the Gulf Stream representation compared to lower resolution models. The mean Gulf Stream in GLOB16 compares well with OSCAR in its path and areal structure, although it overshoots the separation latitude by a few degrees (~ 37°N as in ORCA025, Iovino et al. 2016). In GLOB16, the Gulf Stream is slightly narrower and weaker than observations from 65° to 50°W, where OSCAR depicts a uniform flow towards the Grand Banks. GLOB16 adequately captures the North Atlantic Current (NAC) with the flow turning northwestward around the Grand Banks to separate into a zonal branch heading toward the Azores Islands and a branch flowing towards Newfoundland. At medium- and low-resolution, the Gulf Stream flow is more zonal and significantly weaker - it does not penetrate far into the interior and the recirculating gyre (Fig. 10 c, d, left column). In the ORCA1 ocean, as in many ocean components of the CMIP climate models, the Gulf Stream stays confined west of the New England Seamounts (e.g. Tsujino et al., 2020) with a poor representation of the NAC. The bifurcation into two branches is not correctly reproduced in the eddy-permitting ocean and absent in the non-eddying case, leading to the fresh and cold bias in the Labrador Sea and northwestern subpolar gyre (Fig. 4).

Figure 10 (right column) shows the surface ocean circulation in the North Pacific sector that includes the Kuroshio Current. OSCAR shows that, past Taiwan at ~24°N, the current enters the East China Sea and closely follows the steep continental slope toward Japan, then separates from the boundary approximately at 140°E and 35°N to flow eastward into the open basin of the North Pacific Ocean as the Kuroshio Extension (Kawai, 1972). As for the Gulf Stream, the GLOB16 current presents a clear improvement in reproducing this WBC system – the Kuroshio has similar structure to the OSCAR estimate but is narrower and the separation is shifted northward by about 2° of latitude. The GLOB16 Kuroshio extension, magnitude and its eastward decay match observations with the current reaching 170°E with a ~35cm/sec speed. ORCA025 has a reasonable spatial distribution and amplitude toward 145°E, but decays rapidly further east. ORCA1 substantially underestimates the WBC and its extension (Tseng et al., 2016), with the velocity of the Kuroshio Extension generally lower than 15 cm/sec.

It is worth mentioning also that, previous studies (e.g., Chassignet and Xu, 2017, Ajayi et al., 2020) showed that a prerequisite for significantly intensifying the WBCs and improving the realism of their separation and eastward penetration is to resolve sub-mesoscale activities (with horizontal resolution up to 1/50°).

**Figure 10. Ocean current (in cm s⁻¹) averaged between 0-30m in the Gulf Stream and Kuroshio regions for the 3 simulations and the OSCARv3 dataset, averaged in the last decade 2009-2018.**

Figure 11 shows the complex ocean circulation in the Southern Ocean sector dominated by the ACC and its distinct structure with energetic mesoscales and multiple jets (Ivchenko et al., 2008). Being dependent on mesoscale eddy activity, the ACC structure and intensity are sensitive to ocean model resolution and configuration (Farneti et al., 2015). Even though all models depict the major circulation pattern, the spatial structure and strength in the GLOB16 ocean are in much closer agreement with the OSCAR dataset, following the observed irregular width and pathway. In the Indian Ocean, the Agulhas Current in GLOB16

properly follows observations with the flow down the Mozambique channel and the eastern Madagascar coast that continues along the coast of southern Africa. The Agulhas Current retroflects at the southern tip of the African continental shelf to flow both west into the South Atlantic and east along the Agulhas Return Current. The ACC travels across the Indian Ocean where its southern extreme approaches 70°S, its maxima are approximately at ~45°S. Toward the Pacific sector, the flow passes around and through gaps in Macquarie Ridge and then moves northeast along and around the eastern edge of the Campbell

Plateau (south of New Zealand). In the South Pacific the current is bounded at 40°S and its extension toward Antarctica is limited by the well captured gyre in the Ross Sea. The flow weakens eastward due to the influence of Drake Passage and then extends in the Atlantic Ocean. Downstream of Drake Passage, GLOB16 accurately reproduces the ACC northern branch that breaks off as the Malvinas current and flows northward along the edge of the Patagonian shelf. In the southwestern Argentine Basin, the eddy-driven Zapiola anticyclone is well-placed between the 40°-50°S and its spatial structure and strength are in

close agreement with OSCAR field. The Antarctic coastal current is also clearly represented, it flows westward along the Antarctic coast and meets the eastward-flowing ACC at the Drake Passage. Then, the flow resumes its eastward course across the Atlantic Ocean, where it extends southward to ~60°S with a proper Weddell Sea gyre, and northward between latitudes 40°S and 50°S. It is worth mentioning that the satellite dataset might misrepresent or be less accurate close to the Antarctic coastline or ice-covered areas.

ORCA025 successfully captures most of the circulation features and the circulation pattern agrees reasonably well with observations and the eddying ocean but with reduced amplitude, while ORCA1 struggles to accurately reproduce the main Southern Ocean processes that influence the large-scale ocean circulation. The low-resolution ACC is everywhere weak (generally below 20 cm s$^{-1}$) compared to OSCAR.

0-30m Mean Speed [cm s⁻¹] [2009-2018]

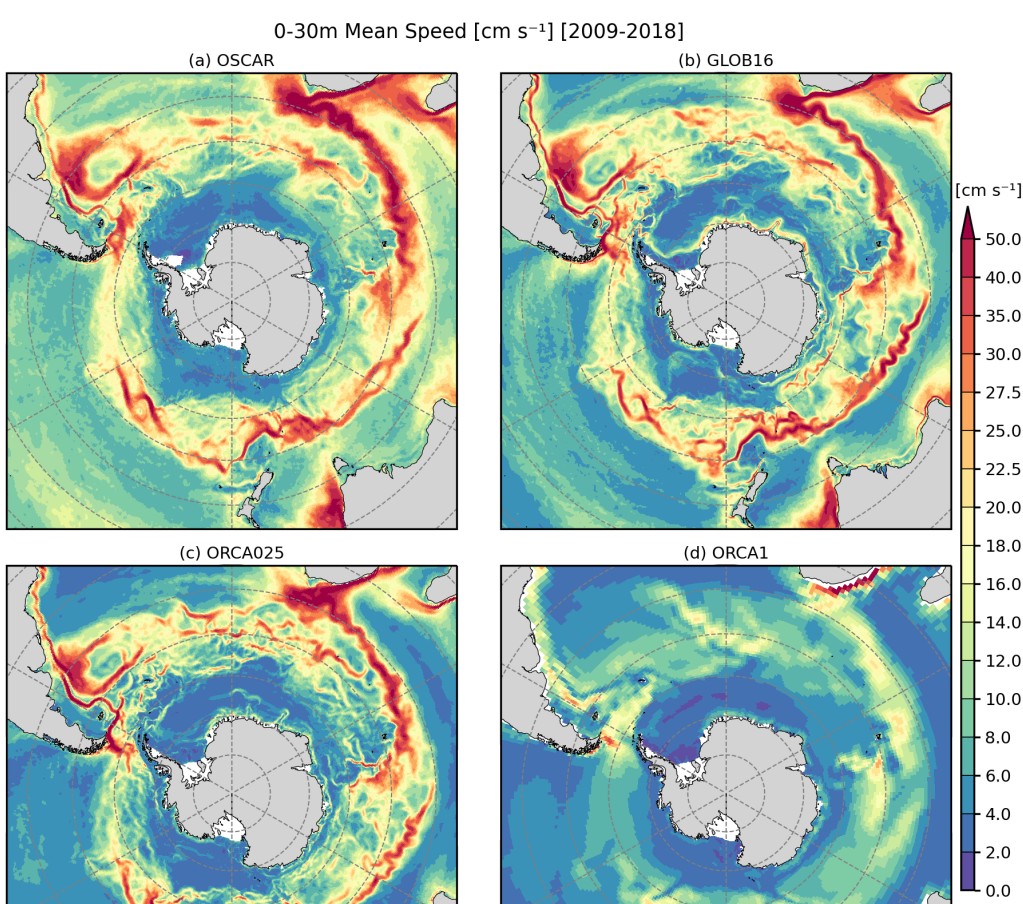

**Figure 11. Ocean current (in cm s⁻¹) averaged between 0-30m in the Antarctic region for the 3 simulations and the OSCARv3 dataset, averaged over the last decade 2009-2018.**

### 3.3.2 Volume and heat transports

Transports of mass, in particular the meridional overturning circulation (MOC), are frequently used to evaluate the model performance. To provide an overview of the large-scale general circulation of the GLOB16 configuration, the meridional overturning stream function is computed for a zonally averaged view. To represent the transport of tracers and quantify the transformation of water masses in different density classes, the calculation is made in density space, and the MOC is shown as a function of potential density referenced to 2000 dbar ($\sigma_2$) from monthly meridional velocity and density fields (see Farneti et al. 2015). The difference between transport in depth versus density coordinates is relevant at high latitudes where isopycnals slope dramatically (Johnson et al. 2019): the differences are due to the (horizontal) transport affected by the subpolar gyre in the North Atlantic, while they arise from the large contribution by mesoscale eddies and standing waves to the transport of density in the Antarctic circumpolar sector.

Figure 12 shows the zonally integrated overturning stream function over all longitudes in the Southern Ocean (south of 30°S) and in the Atlantic Ocean (AMOC, north of 30°S), averaged over the last 10 years of integration (2009–2018) for each of the three model cases (note that the ORCA1 meridional velocity is the sum of the Eulerian-mean velocity and the GM eddy-induced component obtained through eddy parameterization). In GLOB16, the structure of the MOC in the Southern Ocean, from the southernmost boundary to 30°S, agrees well with previous studies (e.g. Farneti et al. 2015). The wind-driven subtropical cell is part of the horizontal subtropical gyres and is confined to the lightest density classes. This anti-clockwise cell comprises a surface flow spreading poleward to 40°S, compensated by an equatorward return flow. Below, the upper cell






is depicted by the large clockwise circulation, which mainly consists of upper circumpolar deep water. The anticlockwise abyssal cell, in the densest layers, occupies a small part of density space but comprises a significant fraction of global water volume. It consists of the poleward lower circumpolar deep water and the deeper equatorward Antarctic Bottom Water (AABW). This abyssal cell is mainly driven by processes of surface water mass transformation over the Antarctic continental shelf; its observed strength is ~21±6 Sv (Ganachaud and Wunsch, 2000). A portion of this overturning cell (~10 Sv) is exported out of the Southern Ocean across 30°S, in agreement to the Southern Ocean State Estimate by Mazloff et al. (2010). While the equatorial-ward lower cell transport is similar in our models, Farneti et al. (2015) showed that low-resolution simulations generally reproduce a weak bottom overturning cell compensated by a strong upper cell. From 60ºS to the Antarctic continent, the transport represents the contribution of subpolar gyres in the Weddell and Ross Seas. After one cycle of JRA-do forcing, it reaches ~15 Sv around 65°S in all runs and is centered at 1036.8 kg m$^{-3}$ in GLOB16 (Fig. 12a) while it is denser in the lower-resolution models (Fig. 12b, c). While the upper ocean takes decades to achieve equilibrium, the deep ocean adjustment requires hundreds of years to reach a quasi-equilibrium state (e.g. Danabasoglu et al., 1996) because of the slow diffusion of active tracers. Tsujino et al. (2020) show that OMIP2 low-resolution simulations take about four cycles to spin-up, and the AMOC declines in the first cycle and slowly recovers thereafter. A longer GLOB16 integration would be necessary to reach a quasi-equilibrium behavior of the overturning in the deep ocean and analyze the long-term evolution of deep-water properties from the initial state also in the eddying ocean. Northward, we present only the Atlantic component that dominates the interhemispheric upper overturning cell at global scales. The AMOC consists of a positive upper/mid-depth cell whose northward branch transports thermocline and intermediate waters and whose southward branch transports North Atlantic Deep Water (NADW), and an abyssal cell associated with the AABW formed in the Southern Hemisphere; at high latitudes, the AMOC involves the sinking of dense water in the subpolar gyre, which upwells at the surface in the Southern Ocean (Marshall and Speer, 2012). In GLOB16, the NADW starts to sink north of 45°N with the maximum transport located at 55°N and the largest densification north of 60°N; the density of the southward NADW flow ranges between 1036.5-1037 kg m$^{-3}$ (corresponding to depth of 1500-3000 m, not shown). About 6 Sv travel northward across the Greenland–Scotland Ridge. The cross-equatorial transport is below 14 Sv. Decreasing resolutions, the overall structure of the transport in the Atlantic Ocean does not change significantly, the magnitude of the overturning south of the Greenland-Scotland Ridge is similar but the density of the sinking water slightly increases, the transport tends to weaken and be restricted in a smaller latitude band. North of the ridge, the transport weakens by ~30% and ~60% in ORCA025 and ORCA1 respectively, suggesting also a reduction of heat supply to the Arctic basin (not shown). In all models, the abyssal cell fills the deep ocean of water denser than 1037 kg m$^{-3}$ and reaches up to 12 Sv in density space or below 3000 m in depth space (not shown).

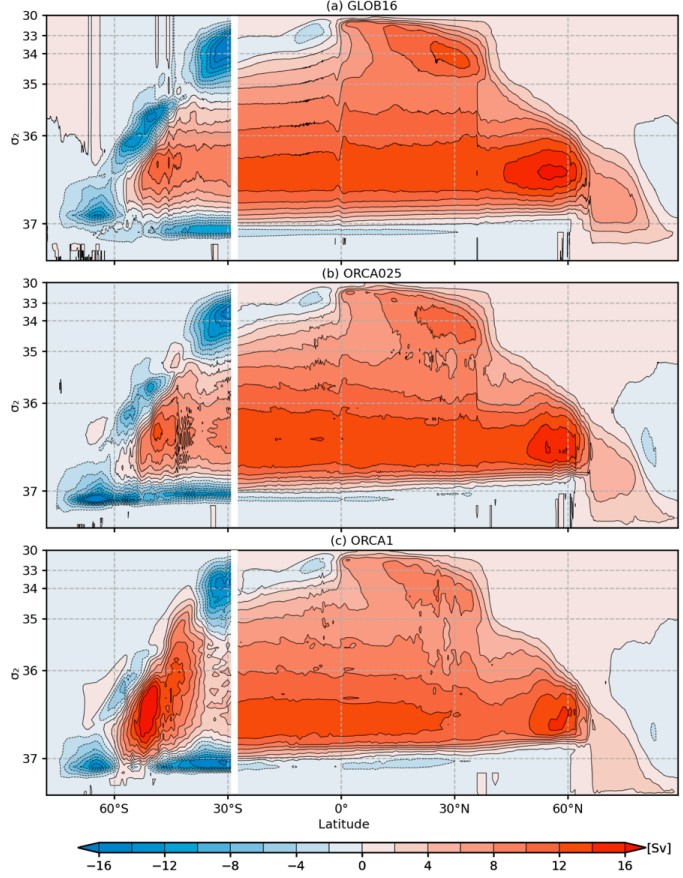

**Figure 12. Time-mean zonally-integrated overturning circulation (in Sv) over the Atlantic sector as a function of latitude and $\sigma_2$ averaged over the period 2009-2018 for the three simulations. South of 30°S, the integral is taken over all longitudes. The density axis is non-uniform, the contour interval is 2 Sv. Positive (negative) stream function indicates strength in the clockwise (anti-clockwise) direction.**

The AMOC in depth and density spaces has specific characteristics due to differences in the zonal integration along a constant density versus along a constant depth surface (e.g Kwon and Frankignoul 2014). While the AMOC in depth space emphasizes changes of isopycnal depth with latitude, the AMOC in density space better represents the transformation of water mass properties with latitude. The differences between the two calculations are hence significant in the Atlantic subpolar gyre, which is characterized by a large density contrast between the warm and salty water from the North Atlantic Current flowing northeastward in the eastern gyre and the return denser (colder and fresher) flow moving southward in the western sector (Hirschi et al., 2020). The maximum values of AMOC as function of latitude are shown in Figure 13 for both calculations, as computed from GLOB16 and ORCA1 models (ORCA025 lies close to GLOB16 – not shown). In both configurations, differences between AMOC in density and depth space are negligible in the Southern Ocean and tropical band, then the two curves start to diverge northwards. In GLOB16, the AMOC in depth coordinate weakens markedly north of about 35°N and declines by 80% north of the Greenland Scotland Ridge (~65°N); the AMOC in density coordinate increases until 60°N, with the highest values (larger than ~16.5 Sv) between about 50°N and 60°N, where it is more than twice as strong as the AMOC in depth. This difference is less pronounced in our low-resolution configuration with a larger fraction of the sinking occurring at the northernmost latitudes, in agreement with previous studies (Zhang, 2010, Danabasoglu et al., 2014, Hirschi et al., 2020).

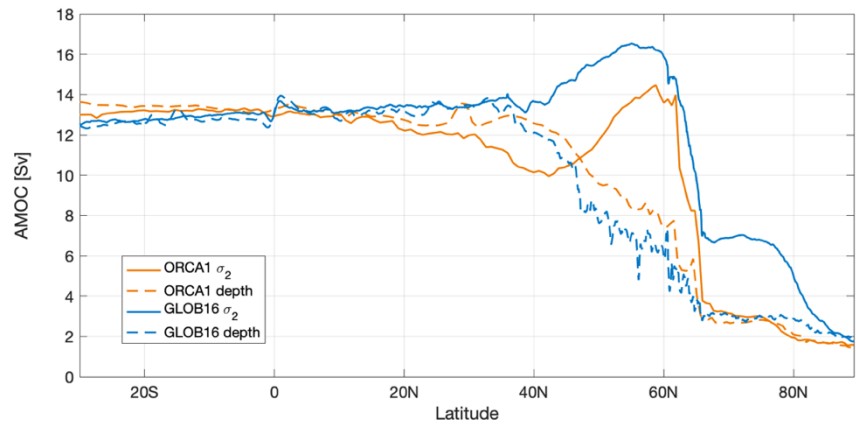

**Figure 13. Maximum values of AMOC (in Sv) in depth (dashed line) and σ₂ (solid line) averaged over the period 2009-2018 for GLOB16 (blue) and ORCA1 (orange).**

The continuously varying strength of the AMOC has been measured across fixed sections at several latitudes, for example at 26.5°N (since spring 2004) and 34.5°S (since 2009). In the former, the magnitude of the AMOC is defined as the maximum of the stream function in depth and represents the total northward transport above the overturning depth. It is made available by the RAPID/MOCHA program (https://rapid.ac.uk/rapidmoc/, Smeed et al., 2018). We compare the time series of the strength of the AMOC at 26.5°N from the eddying-model integration and the RAPID estimates in Figure 14a. Compared to the mean observed value of $16.9 \pm 3.44$ Sv for the period 2005-2018, the modelled AMOC transport is slightly weaker reaching a mean value of 13.6 Sv (OMIP2 high-res simulations range from 14 to ~20 Sv in Chassignet et al., 2020). As other OMIP runs, GLOB16 shows a transport decrease in the first decade, and quasi-zero tendency thereafter to follow the RAPID interannual variability in the last decade. The GLOB16 captures the weak AMOC events observed in 2010, 2011 and 2013. There are no evident changes to the AMOC strength at 26.5°N due to grid resolution (with a mean value of 13.45 Sv and 13.59 Sv in ORCA025 and ORCA1, respectively). Much of the variability at that latitude on interannual timescales is dominated by wind forcing (Pillar et al. 2016), against the previous hypothesis that AMOC variations are driven by the buoyancy forcing in subpolar regions (Kuhlbrodt et al. 2007). All simulations are forced by the same atmospheric reanalysis over a single JRA-do cycle and present similar interannual variability (not shown).

The South Atlantic meridional gap between Africa and Antarctica provides a crossroad for ACC water masses and water masses exchanged between the subtropical Indian and South Atlantic gyres (Speich et al., 2006). The AMOC transport in the Southern Atlantic (Fig. 14b) is estimated on direct daily measurements at 34.5°S from the South Atlantic MOC Basin-wide Array (SAMBA, Meinen et al., 2013), which has a pilot array in 2009-2010 and a second record from 2013 to 2017. It is worth noting that the SAMBA calculation method uses a time-mean reference velocity, so the observations at 34.5°S provide the time-variability of the AMOC rather than an observational mean. The observations yield a peak-to-peak range of 54.6 Sv on daily means, about 20 Sv on monthly means. The AMOC has a time mean meridional transport over the full 2009–2017 period (keeping in mind the ~3-year gap) of $14.7 \pm 8.3$ Sv. Time mean AMOC transport in GLOB16 is 12.1 Sv over the same period, with a weaker interannual variability. Transport in the medium- and low-resolution oceans compares in magnitude (13.8 Sv in both runs) and time variability at interannual and decadal scales with GLOB16 (not shown).

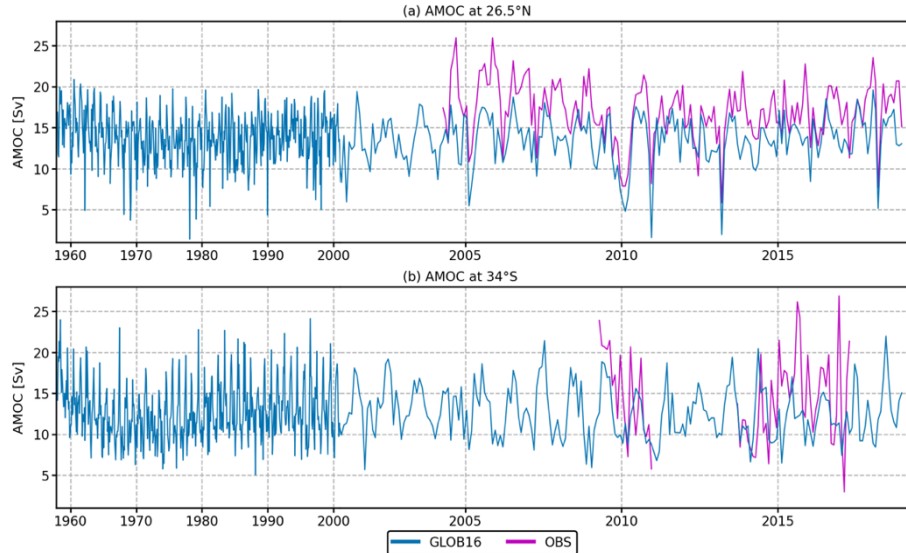

**Figure 14. Time evolution of monthly mean AMOC transports, defined as the maximum value of the global overturning stream function in GLOB16 (blue line) computed (a) across 26.5°N and compared to RAPID estimates, and (b) 34°S compared to SAMBA record. Scale is compressed prior to year 2000.**

The mean Atlantic meridional heat transport (AMHT) averaged over the last 10 years of integration is presented as function of latitude in Figure 15a, as reproduced by the three models, in comparison with a suite of direct and indirect observational estimates. The range of observed transports is quite broad: the location of heat transport maximum and its magnitude are observation dependent. The AMHT peak is close to 22°N in the estimates by Large and Yeager (2009, LY09 in figure), and around 18°N in the European Centre for Medium-Range Weather Forecasts (ECMWF) estimates by Trenberth and Caron (2001, TC01). The maximum is widely extended between 20°- 30°N in the Trenberth and Fasullo (2008, TF08) and between 10° - 20°N in the more recent estimates derived from JRA55-do (Tsujino et al., 2020). In the direct measurements by Ganachaud and Wunsch (2003, GM03), the AMHT reaches a maximum of 1.27 at 24°N, with an error bar of ± 0.3. Direct measurements are the largest estimates, followed by LY09 and JRA55 at all latitudes, and up to ~25% larger than estimates from ECMWF reanalysis (TC01) and TF08. All model configurations reproduce the large-scale features and latitudinal variation of the observed profiles, with the Atlantic Ocean carrying heat northward (positive transport) at all latitudes. Models underestimate the mean heat transport relative to in situ measurements, LY09 and JRA55 reanalyses, as also seen in the OMIP and COREII coarse resolution models (Tsujino et al., 2020, Danabasoglu et al., 2014) and in the eddy-permitting and eddy-rich ocean and climate models (Chassignet et al., 2020, Griffies et al., 2015, Msadek et al., 2013). Finer ocean resolution leads to increased heat transport in the northern hemisphere and brings GLOB16 in an overall better agreement with observations. GLOB16 tracks the ECMWF estimates and compares well with TF08 south of 40°N. The three simulations are similar in the Southern Ocean with the heat transport ranging within 0.1 PW, while the mean North Atlantic heat transport is always higher in the eddy-rich ocean than eddy-permitting and lowest resolution models (Chassignet et al. 2020, Hirschi et al. 2020). The GLOB16 maximum heat transport of about 0.88 PW is located at ~25°N. The maxima are not collocated in latitude in the two other models: the meridional distribution of ORCA025 heat transport is very close to GLOB16 in the North Atlantic, with the largest value (~0.78 PW) distributed over a wide band of latitudes between 5° and 30°N, while the AMHT in the non-eddying model presents a peak of ~0.75 PW at ~14°N to rapidly drop toward 45°N and increase again between 45° and 55°N with a marked positive slope that indicates a gain of heat in the subpolar gyre. This simulated increase of heat transport at high latitudes reflects insufficient heat loss to the atmosphere between mid- and subpolar latitudes; it is present in ORCA025 too and many coarse and eddy-permitting models (e.g. Danabasoglu et al., 2014; Grist et al., 2010, Petersen et al., 2019), it is less pronounced in GLOB16, likely due to a correct path of the simulated North Atlantic Current (e.g. Treguier et al., 2012, Robert et al., 2016).

We also assess the distinct contribution of the overturning and horizontal gyre circulations (Fig. 15a) to GLOB16 ocean heat transport. Following Johns et al. (2011), the total AMHT is decomposed into vertical and horizontal heat transports, assumed to represent "overturning" or "gyre" heat transports, respectively. The overturning dominates the AMHT over a large latitude range (e.g., Msadek et al., 2013; Xu et al., 2016), slightly exceeds the total AMHT between the equator and 15°N and south of 20°S, where the gyre component is weakly southward, decreasing the total northward heat transport. At 26.5°N, the breakdown into the overturning and gyre transports agrees well with RAPID observations: the gyre circulation accounts only for slightly more than 10% of the total AMHT (McCarthy et al., 2015). Northward, the overturning component drops and the gyre component increases to level off at total AMHT at 42°N, from there on the horizontal circulation dominates the Atlantic heat transport and explains the large GLOB16 MHT compared to observed TC01 and TF08 (in agreement with the eddying climate models by Griffies et al., 2015). In the eddy-permitting simulation, the overturning and gyre components follow the GLOB16 ones at all latitudes, while, in the non-eddying simulation, the gyre component ranges between ±0.1 PW from the equator to 40°N and then rapidly increases, becoming dominant north of 47°N (not shown).

At 26.5°N, the AMHT is significantly smaller than the observational estimates at 26.5°N in all cases (Fig. 15a). GLOB16 generally underestimates the mean RAPID value that equals 1.14 PW with an error bar of ± 0.032 (Bryden et al. 2020), as well as the RAPID estimates all through the RAPID record (Fig. 15a, b). Similar behavior can be seen in many model studies covering a large range of horizontal resolution (e.g. Maltrud and McClean, 2005; Mo and Yu, 2012; Danabasoglu et al., 2014). GLOB16 misrepresents the interannual variability in the first ~5 years of RAPID record to better follow the data variability onward capturing the 2010, 2011, 2013 and 2017 minima. It is worth mentioning that several studies (e.g. Sinha et al. 2017, Roberts et al. 2013, McCarthy et al. 2012, McCarthy et al. 2015) have discussed the potential for structural errors associated to the measurement design and calculation methodology of the RAPID basin-wide estimates. Among those, Stepanov et al. (2016) provided insight into understanding the source of dissimilarities between the Atlantic heat transport at 26.5°N as simulated in ocean models (in the GLOB16 eddy-rich and ORCA025 eddy-permitting regimes) and estimated from the RAPID array. They quantified how the values of AMHT depend on the calculation method, in particular the RAPID-like calculation (following Johns et al., 2011) applied to model outputs was compared to the classical model calculation using 3D output of temperature and velocity fields (model truth). They found that the negative AMHT bias generally obtained from models can be directly linked to the applied calculation method rather than a potential weakness of the model itself in reproducing the observed transports. In their study, the RAPID-like calculation leads to an AMHT increase of about 20% that can at least partially explain the discrepancies between the true model AMHT and the RAPID estimates.

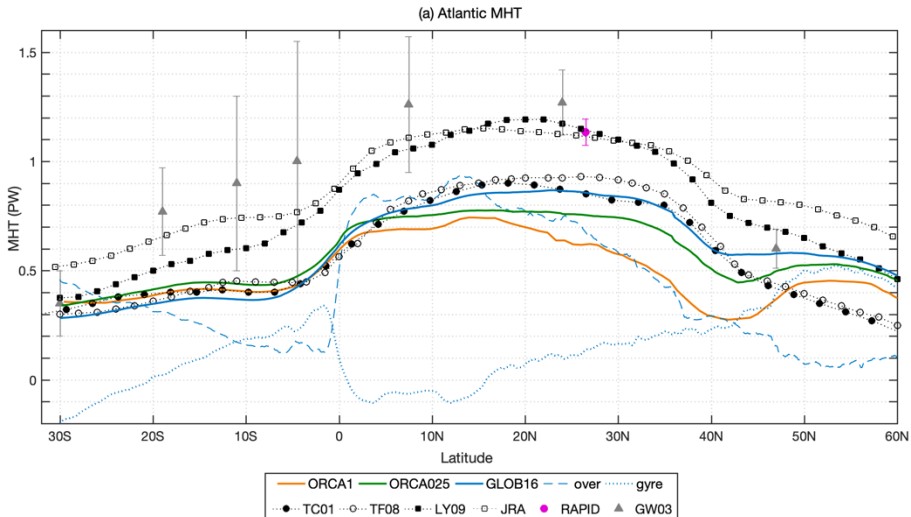

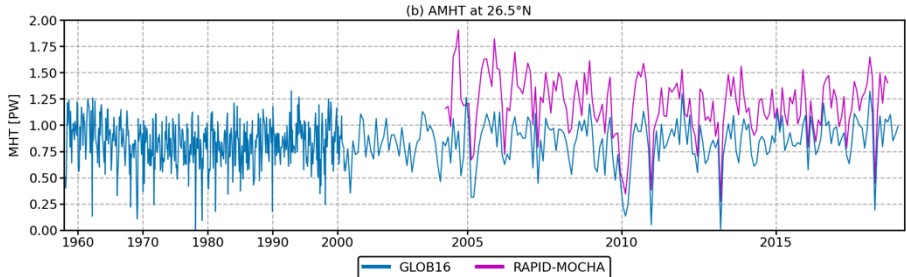

**Figure 15. (a) Atlantic meridional heat transport (in PW, positive northward) in ORCA1 (orange), ORCA025 (green) and GLOB16 (blue, divided in its overturning (dashed) and gyre (dotted) components), averaged in 2009-2018, compared with direct and indirect observational estimates. TC01 corresponds to ECMWF estimates by Trenberth and Caron (2001), TF08 to Trenberth and Fasullo (2008), LY09 to Large and Yeager (2009), JRA refers to JRA55-do v3 estimates in Tsujino et al. (2020). GW03 and RAPID refer to Ganachaud and Wunsch (2003) and RAPID array, respectively. The vertical bars indicate the uncertainty range for the direct estimates. (b) Times series of the monthly-mean total AMHT in GLOB16 (blue line) across 26.5°, compared to the RAPID record (magenta). Scale is compressed prior to year 2000.**

The strengths of the GLOB16 volume transports across key passages agree well with observations and are generally within or very close to the limits of observed uncertainty. The simulated Pacific inflow across the Bering Strait (Fig. 16a) tends to be slightly large in GLOB16 compared to lower resolution. During the first two decades where observations are available, GLOB16 overestimates the recent estimates by Woodgate and Peralta-Ferriz (2021), it cannot depict the increasing northward flow (0.01 ± 0.006 Sv/year), but it follows very closely the observed interannual variability in the last simulated decade from the ~0.8 Sv minimum in 2010 (Woodgate 2018). The large transport at Bering Strait is common to many NEMO simulations and does not depend on the grid resolution (e.g. Marzocchi et al. 2015).

The total Indonesian Throughflow (ITF, negative transport into the Indian Ocean) measures water exchanges between the Pacific and Indian Ocean. Water masses that flow through the ITF are advected westward to feed the upper limb of the meridional overturning circulation in the Southern Atlantic Ocean and contribute to the Agulhas Current. The volume transport estimates from the INSTANT Program over a ~3-year period during 2004–2006, corresponds to 15.0 Sv, varying from 10.7 to 18.7 Sv (Sprintall et al., 2009; Gordon et al., 2010). The GLOB16 ITF transport (in Fig. 16b) is computed between Indonesia and Australia across the three outflow passages of Lombok, Ombai, and Timor straits. It falls within the range of minimum and maximum values from INSTANT but slightly underestimates the observed mean value. While the ITF has no evident drift in the first 20 years, it exhibits a gradual decrease afterwards with large interannual variability. The differences among models are impacted by the model accuracy in realistically representing ocean topographic features, such as narrow straits. At lower resolutions, the total transport has smaller or no evident drift over time and is generally above the mean observed value. The effect of resolution on the interannual variability is small.

Figure 16(c) presents the time series of the annual mean Drake Passage (positive eastward) that is representative of the large-scale features and strength of the ACC where it is constricted between the Antarctic Peninsula and the southern tip of South America. In the Southern Ocean, low-frequency adjustment to local and remote forcing and deep bottom water formation processes likely require longer integrations for stabilizing the ACC transport – also coarse models may still present significant trends and have not reached an equilibrium after the fifth cycle of atmospheric forcing (Farneti et al., 2015). Substantial efforts have been made toward measuring ACC transport, especially in the Drake Passage from the late 70s. Mean observed values of the full-column transports range from mean strength of 127.7±8.1 Sv (Chidichimo et al., 2014), to 129±6 Sv from the World Ocean Circulation Experiment hydrographic data (Lumpkin and Speer, 2007), to 134±13 Sv based on the International Southern Ocean Studies (ISOS) program (Whitworth and Peterson 1985), to 135.3±10.2 Sv based on hydrography cruises from 1993 to 2020 along the line SR1b (e.g. Xu et al., 2020), to 141±2.7 Sv and 173.3±10.7 Sv based on the DRAKE (Koenig et al., 2014) and cDRAKE (Donohue et al., 2016) programs, respectively. The GLOB16 time series of the yearly averaged transport shows a fast decline in the first 20 simulated years, then the drift becomes negligible, and the transport stabilizes at

a level of about 100 Sv, below the most recent estimates (Xu et al., 2020) and some eddy-rich models (kiss et al., 2020). The eddy-permitting ocean presents a similar behavior with a smaller decrease and ~120 Sv at the end of the integration. As already shown in the CORE-II intercomparison, the mean transport at the Drake Passage is generally larger than observational estimates in non-eddying oceans (Farneti et al., 2015). This is confirmed by our low-resolution transport that is indeed above observations, the time series presents a smaller decrease and levels off at 150 Sv, comparable to the mean transport of ~160 Sv from the low-resolution OMIP2 runs in the first JRA cycle (Tsujino et al. 2020, Chassignet et al., 2020). The simulation of the ACC is sensitive to the grid resolution in both forced and coupled simulations (Hewitt et al., 2020). In contrast to the eddy-permitting and eddy-rich ocean models, the non-eddying regime fails to represent distinct ACC frontal jets (Beadling et al., 2020), the time-mean flow across the Drake Passage is all eastward and there is no evidence of small intermittent westward currents. In the higher resolution models, the ACC structure agrees with the observed frontal locations and intensity, and the time-mean velocity is characterized by distinct counter flows, possibly linked to stationary mesoscale features that may not be evident in long-term observational means over different periods (Hewitt et al. 2020). This feature can partially explain the reduced ACC transport in eddying oceans. Variability in ACC strength is shown, by observations and models, to be relatively insensitive to atmospheric forcing changes due to the eddy saturation (Hallberg and Gnanadesikan 2006): additional energy imparted from the winds is cascaded to the oceanic mesoscale instead of inducing prolonged accelerations of the horizontal mean flow. The net overturning is determined by a balance between a wind-driven circulation and an opposing eddy-induced transport.

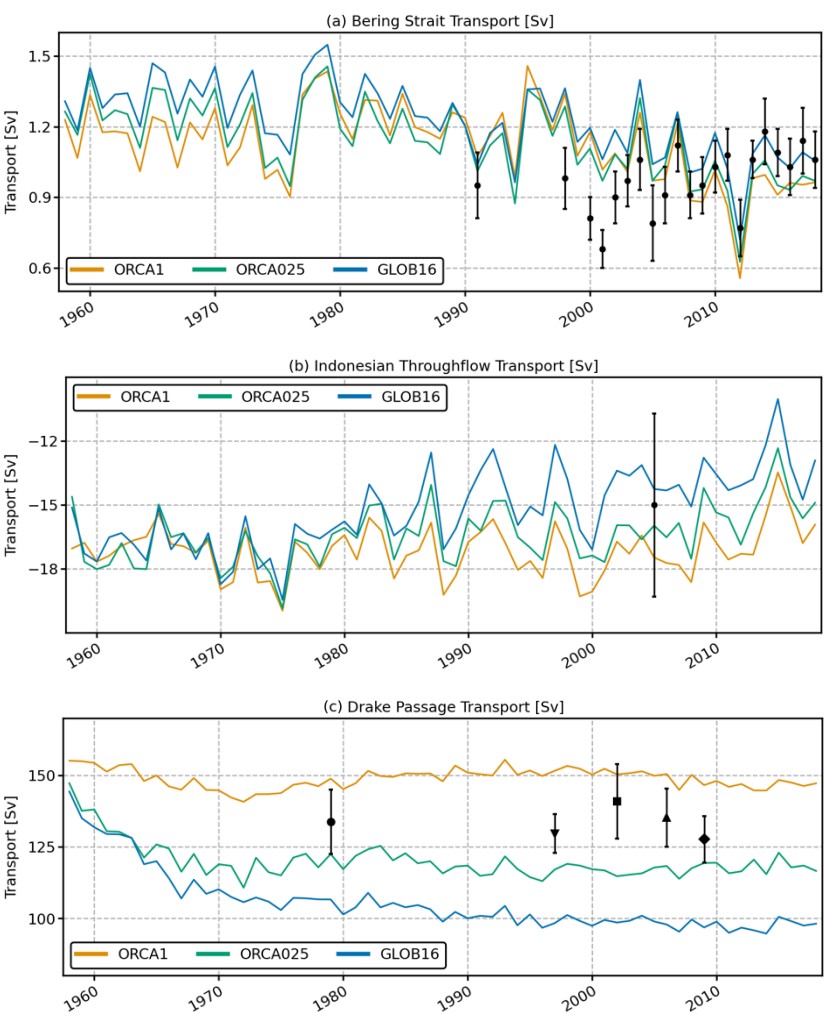

**Figure 16. Time evolution of annual-mean volume transport (in Sv) of (a) the northward flow through the Bering Strait, (b) the Indonesian Throughflow from the Pacific to the Indian Ocean, and (c) the ACC through the Drake Passage. Observed values with error bars are shown. Estimates from Woodgate and Peralta-Ferriz (2021) and INSTANT Program (Sprintall et al., 2009) are shown for the Bering Strait and ITF transports, respectively. A suite of observations is shown for the ACC transport: Whitworth (1983),**

**Whitworth and Peterson (1985) (circle), Lumpkin and Speer (2007) (triangle down), Koenig et al. (2014) (square), Xu et al. (2020) (triangle up) and Chidichimo et al. (2014) (diamond).**

## 4. Sea ice

Formation and melting of sea ice strongly affect the ocean dynamics both locally in polar regions and in the global ocean, through the influence on exchanges between atmosphere and ocean and the contribution of high-latitude processes in deep water production. Changes in sea ice can greatly affect ocean hydrography, ocean dynamics and heat transport. There are two sea ice models used in this study that have large differences in their complexity and their default sea ice initial conditions. However, a detailed analysis of the impact of sea ice model complexity and sources of simulated sea ice differences is beyond

the scope of the present study.

    Here, we present sea ice cover and its variability for both hemispheres as simulated by the three numerical experiments in comparison with satellite observations, over the period 2009– 2018. Sea ice extent is defined as the area of the ocean with an ice concentration of at least 15%. The climatological mean seasonal cycle of the Arctic and Antarctic sea ice extent (SIE) as reproduced by the three model configurations is shown in Figure 17, together with estimates from two satellite-based products,

the NOAA/NSIDC Climate Data Record v. 4 (Meier et al. 2023, https://nsidc.org/data/g02202/versions/4) and the EUMETSAT Ocean and Sea Ice Satellite Application Facility Climate Data Record v. 3 (OSISAF 2022, https://navigator.eumetsat.int/product/EO:EUM:DAT:0826).

    In the Arctic region, the simulated seasonal cycle is consistent among models and in general agreement with observations (Fig. 17a). All simulated and observed products have a maximum in SIE in March and a minimum in September. GLOB16 closely

follow the observed seasonality, except in summer when it shows a weaker decline with an overestimated minimum, $6.2 \times 10^6$ $km^2$ compared to $\sim5 \times 10^6$ $km^2$ from the satellite products. In all configurations, the mean SIE in March is close to the observed SIE ($\sim15 \times 10^6$ $km^2$), but the model spread increases in summer/autumn months when sea ice decline is too quick in the lower-resolution models, resulting in a mean SIE in September about 30% smaller than observations (3.2 and $3.6 \times 10^6$ $km^2$ in ORCA1 and ORCA025, respectively).

In the Southern Hemisphere, the seasonality simulated by the three models is, to a great extent, in good agreement with the observations (Fig. 17b), but all models on average tend to have a weaker seasonal cycle with lower SIE than the observations. All models undervalue the observed amplitude of $\sim16 \times 10^6$ $km^2$, GLOB16 the smallest ($13 \times 10^6$ $km^2$ against $\sim15 \times 10^6$ $km^2$ in both ORCA1 and ORCA025). In GLOB16, the melting process are slower with a smaller sea ice decline in the austral spring and summer. That results in a larger SIE minimum in February. This is explained by an overestimation of sea ice thickness in

austral autumn and winter (not shown), constantly simulated from the beginning of the integration and related to a too large sea ice thickness used to initialize the run. Therefore, more heat is needed to melt sea ice and produce open ocean area. ORCA1 and ORCA025 shows smaller Antarctic SIE than the satellite estimates by $\sim10\%$ year-round.

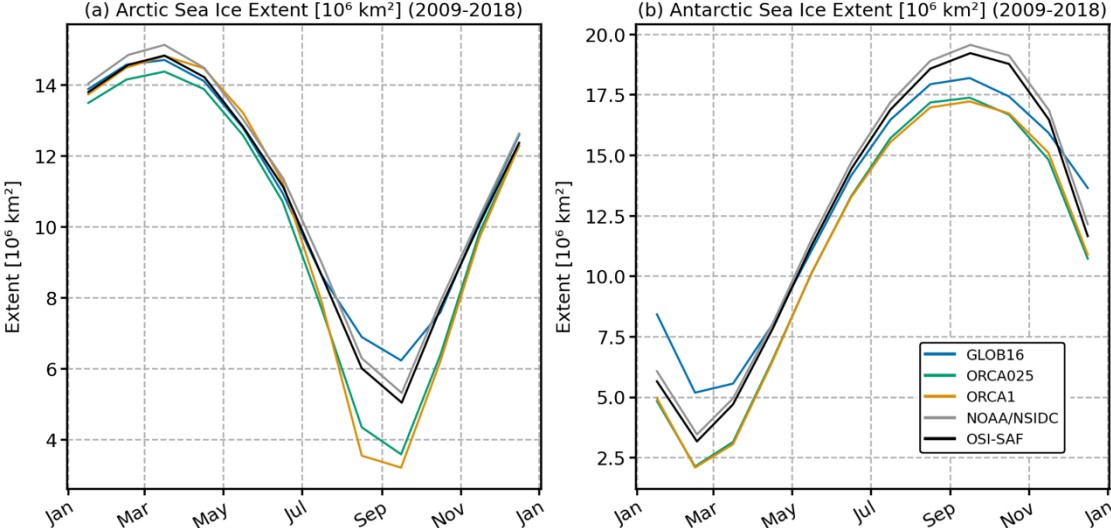

**Figure 17. Mean seasonal cycles of sea ice extent ($10^6$ km²) for the Arctic (a) and Antarctic (b) regions compared to satellite observations provided by NSIDC and OSISAF. Sea ice extent is defined as the area enclosed in the 15% sea ice concentration contour.**

Comparison between the spatial distribution of the simulated sea ice concentration (SIC) and the OSISAF estimates averaged over 2009–2018 shows that the simulated sea ice distribution in the end of the growing seasons is realistic, also in terms of ice edge, in both hemispheres (Figs. 18a-d and 19e-h), although ORCA1 simulates a slightly smaller sea ice coverage around Antarctica. The winter SIC distributions are similar among the models, although ORCA1 and ORCA025 exhibit a slightly excessive ice concentration in the central Arctic (Fig. 18c, d) and all model configurations, in particular GLOB16, tend to underestimate the concentration of the Antarctic consolidated pack ice (SIC > 80%), especially in the Weddell sea (Fig. 19f). The sea ice edge and the ice geographical distribution of the summer minimum concentration are generally well simulated by the models (Figs. 18e-h and 19a-d). In the Northern Hemisphere, GLOB16 concentration is slightly too high in the Beaufort Gyre and extents too far south (to 70°N) in the Arctic Pacific sector (Fig. 18f). In the coarser resolutions, Arctic summer minimum ice cover is smaller than observed; the region covered by pack ice is underestimated with a consequent retreat of the marginal ice zone (15% < SIC < 80%), in agreement with the strong melting and low summer minima (Fig. 17). In the Southern Hemisphere, the spatial distribution of the summer Antarctic sea ice is generally consistent with OSISAF, but the area covered by sea ice is wider at all resolutions (Fig. 19 a-d), with the highest value located close to the Antarctic Peninsula in the Weddell Sea, where the pack ice region is anyway smaller than the observed one. In the Weddell Sea and in the Amundsen and Ross Seas, the GLOB16 ice edge is located far south in the models compared to OSISAF, with a broad region of low concentration that extents to 65°S. In the coarser resolution configurations, with a weak Antarctic coastal current (Fig. 11), the Indian and Western Pacific Ocean are ice-free in summer.


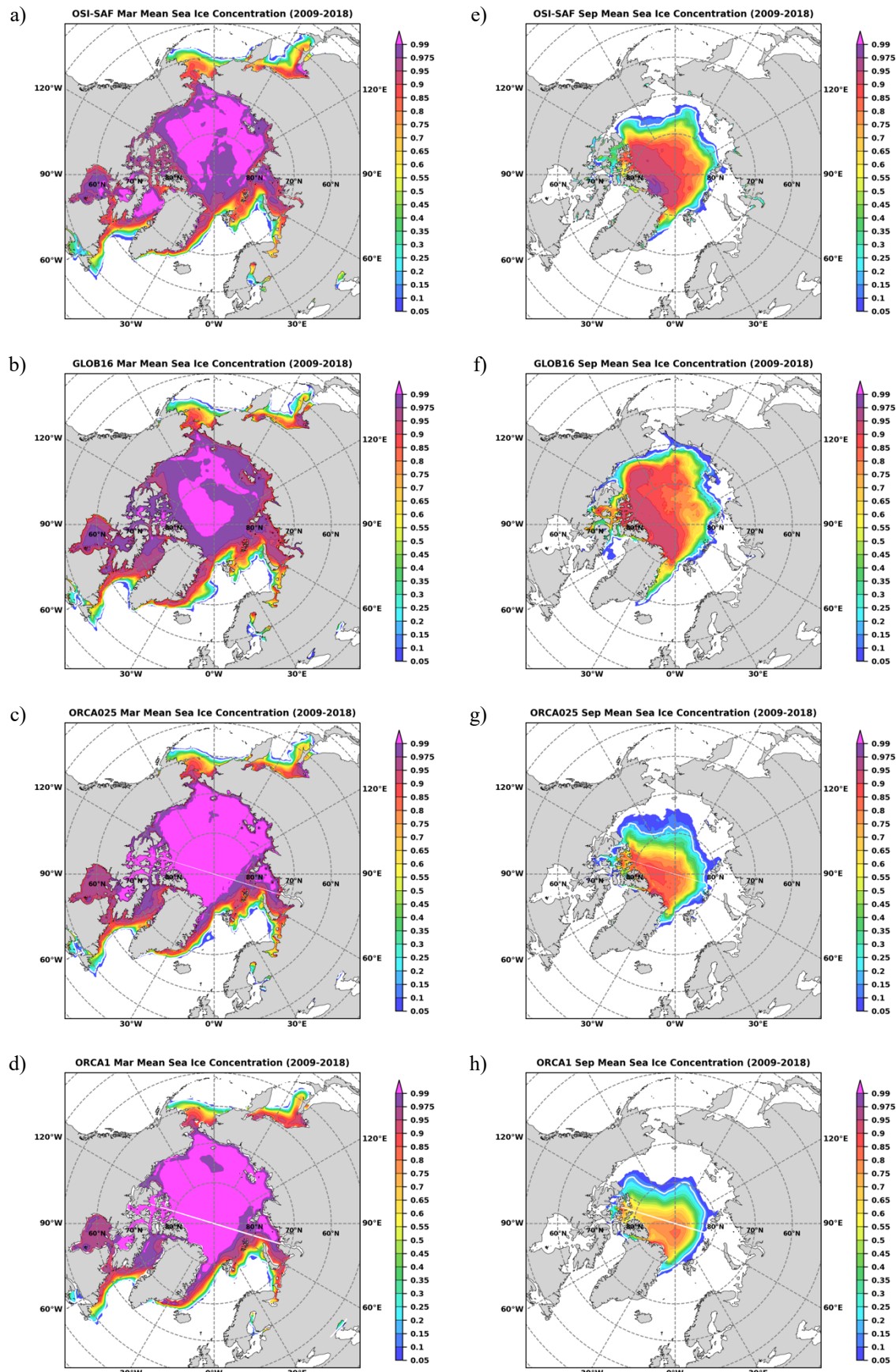

**Figure 18. March (a - d) and September (e - h) climatology of Arctic sea ice concentration (ice area per unit area) for the period 2009–2018 from satellite-based estimates (OSISAF 2022) and in the three model configurations.**


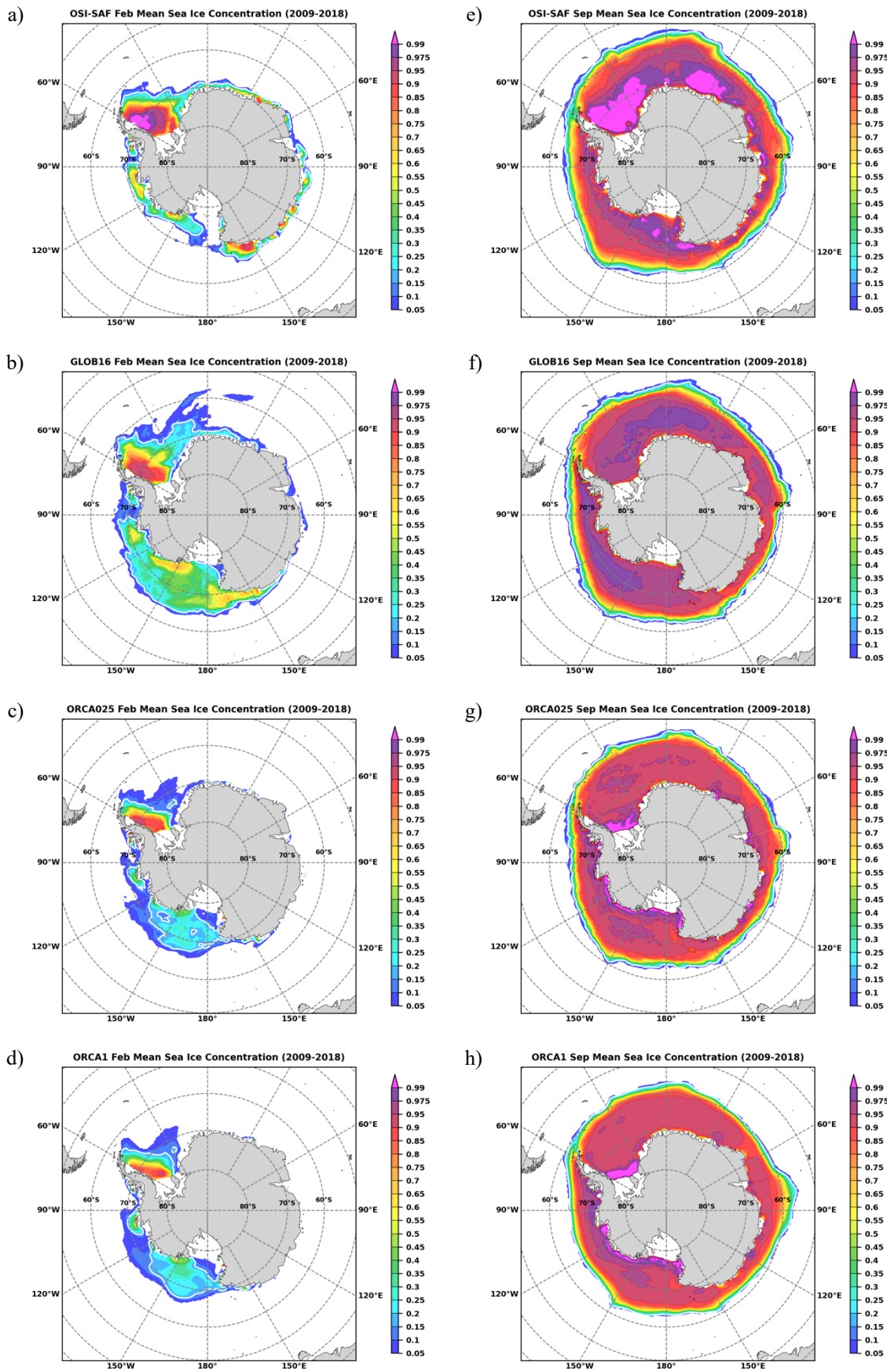

**Figure 19. February (a - d) and September (e - h) climatology of Antarctic sea ice concentration (ice area per unit area) for the period 2009–2018 from satellite-based estimates (OSISAF 2022) and in the three model configurations.**

## 5. Concluding summary

The OMIP2-like simulation performed by the CMCC ocean and sea-ice model at eddying horizontal resolution, GLOB16, is described and evaluated in this study. GLOB16 employs the NEMOv3.6 ocean model coupled to the LIM2 sea ice model. While it is generally applied to perform short-term ocean forecasting for operational purpose, here GLOB16 has been used to perform a longer benchmarking experiment based on the OMIP2 framework. The eddy-permitting and non-eddying ocean-sea ice systems are components of the CMCC-CM2 and CMCC-ESM2 based on the CESM infrastructure, and they use

NEMOv3.6 and CICEv4.1.

Due to their different applications, the CMCC global ocean-sea ice model suite is not specifically designed as a model hierarchy for investigating the sensitivity of ocean solutions to grid resolution. However, all models follow, as close as possible, the OMIP2 experimental and diagnostic framework. Only the low-resolution experiment has been previously evaluated in a complete OMIP2 integration (six JRA55 cycles); Tsujino et al. 2020 showed that it reproduces the ocean-sea ice climate at the

level of realism comparable to results from a majority of the OMIP2 low-resolution models in a wide range of indices.

Goal of this evaluation exercise is to evaluate the GLOB16 model performance and to document if and how the CMCC "eddy-resolving" ocean model resolution change the representation of large-scale ocean variability with respect to observations and lower-resolution models, highlighting the relative advantages and disadvantages of running ocean–sea ice models at such resolution. The analysis highlights a general improvement of many key metrics used in climate modelling when the ocean-sea

ice system is run at eddying resolution. The GLOB16 ocean assessment informs which aspects of the model can be used for climate study and provides a benchmark for future developments. As one might expect, the GLOB16 simulation usually presents better results compared to lower-resolution oceans, this is clearly the case for surface currents and internal variability. We show that additional horizontal resolution does not necessarily improve distinct biases in temperature and salinity in all regions. Because of the relatively short integration time, some of the results, such as deep ocean circulation and overturning

variability, may not be robust yet (Danabasoglu et al., 2016, Chassignet et al., 2020). Overall, the GLOB16 upper ocean mean state and variability are well reproduced when compared to observational records and the gain due to finer resolution is robust when compared to a coarser resolution ocean. Large-scale surface circulation, patterns of western boundary currents, the Gulf Stream behavior and associated North Atlantic SST biases, ocean heat content, mass exchange from the Pacific to the Indian Ocean, and from the Pacific into the Arctic Ocean are all much improved in GLOB16, when resolution is refined.

Several aspects of the ocean dynamics need further process-focused analyses and ocean model development activities, such as the AMOC magnitude and variability, and the weak ACC transport (weaker than observed values). These GLOB16 shortcomings are partly due to the relatively short integration length needed by eddy-rich simulation to accurately resolve the response of the deep ocean.

The GLOB16 improvements and weaknesses presented in this study are consistent with results from the previous

intercomparison of OMIP2 runs carried at low- and high-resolution (Chassignet et al., 2020). In spite of its shortcomings, the evaluation leads us to conclude that GLOB16 appears to be competitive with similar models from other institutions (Chassignet et al. 2020, Kiss et al. 2020, Li et al 2020), and the finer resolution remains one possible way in which model capabilities can be enhanced.

## 6. Code and Data Availability

The NEMO model is freely available and distributed under the CeCILL v2.0 license. The version 3.6 code, that includes the LIM sea ice model, can be downloaded from https://forge.ipsl.jussieu.fr/nemo. The CICE4 code is available through the CICE Consortium GitHub. The ORCA1 and ORCA025 model output is published on the Earth System Grid Federation nodes. The GLOB16 model results presented in this paper are available at https://doi.org/10.5281/zenodo.7752243. The atmospheric forcing for the OMIP-2 exercise is available as input datasets for Model Intercomparison Projects (input4MIPs) at https://esgf-

node.llnl.gov/search/input4mips/ (last access: 16 March 2023). All observed datasets are publicly available at the links provided into the text.

## 7. Author contribution

DI, and SM conceived and designed the experiments. PF performed the simulations. DI and PF analysed the model output and observational data. DI wrote the manuscript with contributions from all co-authors. All authors provided scientific input.

## 8. Competing interests

The authors declare that they have no conflict of interest.

## 9. Acknowledgements

This research was supported by the Foundation Euro-Mediterranean Center on Climate Change (CMCC, Italy) and the European Union's Horizon 2020 research and innovation programme under the grant agreement No. 862923 via the project AtlantECO (Atlantic ECOsystems assessment, forecasting & sustainability). We thank the present and past members of the CLIVAR Ocean Model Development Panel who have designed and supported the Ocean Model Intercomparison Project (OMIP). We acknowledge PRACE for awarding us the project #2016163920 (Understanding the role of mesoscale eddies in the global ocean, ROMEO) and providing access to resource on Marconi KNL based at the Cineca (Italy).

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
