# Peer review of "Intercomparison Project (OMIP2)"

_EGUsphere, 2023_

## Author Response (AR1)

Dear Editor and Reviewer,

We would like to thank you for accurately reading and commenting the manuscript and suggesting how to improve it. Answers to your comments are given in details hereafter. We hope that you will find them satisfactory. All authors agree with the modifications made to the manuscript. Reviewer comments are in black and are followed by our response (in blue) that includes changes and/or additions to the text.

For the authors,
Doroteaciro Iovino

**RC1**: 'Comment on egusphere-2023-469', Anonymous Referee #1

The present manuscript presents a high level overview of the differences and similarities between three configurations of the CMCC ocean model in forced simulations. The comparisons are limited to the simulated state - no of process level explanations of the differences are provided. Further, while the emphasis is on the impact of horizontal resolution, there are a large number of other differences in the models being compared: vertical resolution, topography, the sea ice component model and the sea ice initialization, the salinity restoring timescale, the way that runoff forcing is applied, along with the more typical adjustments to viscous and diffusive parameterizations. So, while the authors have put in a considerable amount of effort in compiling model metrics, the resulting manuscript is rather unsatisfying. We have little new insight into how and why the explicit representation versus parameterization of the mesoscale impacts the simulated ocean state. As a documentation of what was done the manuscript may be adequate with some additional work to correct imprecise descriptions, but I do not envision the manuscript having much of an impact beyond those who might wish to use the CMCC models and it is unlikely to advance the field. Additionally, the manuscript suffers from being poorly prepared with many missing references and poor language constructions.

This study exploited numerical system already implemented and in use and took advantage of ongoing simulations rather than a dedicated set of tests and experiments. The low/medium- and high-resolution should have been configured in tandem but they were designed independently for different scientific goals and developed for distinct applications. So, the models are diverse in their components, numerics and parameterizations and we do agree that the effects of horizontal resolution cannot be completely isolated. We have better addressed this issue in the introduction, model description and conclusions and have explicitly highlighted the need of an improved design of future comparisons.
This work mainly aims to present the performance of the eddy-rich ocean/ice system in relation to the lower resolution configurations in hindcast runs. Specific analysis on how the explicit representation of mesoscale dynamics impacts the ocean state will be shown in dedicated studies for instance on the variability of ocean eddy kinetic energy and the eddy exchanges from narrow boundary currents and basin interior in marginal seas (manuscripts are in preparation).
We closely followed comments and suggestions to improve the quality of the manuscript.

Detailed Comments:

Line 26 "enforces our ability in understanding": poor wording, perhaps is a prerequisite to develop our understanding
We reworded the sentence as follows "An accurate representation of the ocean dynamics within the climate system is crucial to understanding drivers of climate change and variability, and to determining the ocean-ice influence on atmospheric circulation and ecosystems."

Line 29-30: ensemble size is a mother strong trade-off in both prediction and climate modeling.
We reworded this sentence as follows "Despite the ongoing increases in computer power and improvements in techniques, a major challenge in climate model design is the trade-off between the level of model complexity, the length of simulations, the choice of ensemble size and the spatial resolution of different climate components."

Line 31 "start grid spacing": rather, typical grid spacing. No standard (a specification) exists.
Done

Line 33 "miss key processes": it is not necessarily the case that key processes are missed. They may be parameterized.
We reworded this sentence as follows "Both resolutions lack an explicit representation of ocean mesoscale dynamics in most of the global domain."

Line 36 "Despite": While simulations at this resolution …
We modified the sentence as follows "Simulations of the global ocean domain at this resolution still require significant computational resources, which limits the number and length of runs and the capacity to optimize the model setup. However, thanks to the ever-increasing processing and storage capabilities of the supercomputers, running global models capable of resolving mesoscale dynamics has become feasible for climate simulations. It is now necessary to assess to what extent the enhanced resolution translates into an improved ocean state."

Line 38: "access to which": to assess to what extent
Done

Line 60 "resolution dependent": as discussed at top this is not a convergence study in the sense of numerical analysis. Many other aspects of the simulation besides horizontal resolution are change.
Thanks for the comment. We reworded the sentences as follows "We run OMIP-like simulations with the three models driven by the same forcing dataset, and we compare them in order to identify possible climate-relevant improvements in the ocean response as model resolution increases. It is worth mentioning that the models do not differ only in the horizontal resolution and associated physical parameters since the high-resolution simulation was configured independently for distinct scientific applications and followed a specific development strategy."

Line 68: Manral et al 2013 ref missing
Reference added

Line 104: suggest starting new paragraph with sentence beginning "While the best …"
Done

Line 154 "in coupled runs": all the runs described herein are forced not coupled?
We do agree with the referee; this sentence was misleading since all simulations are forced even when the framework of the coupled system is used. We modified the sentence as follows "the initial sea ice properties in ORCA1 and ORCA025 runs are...".

Line 189: do not correspond to what _was_ found in
Done

Line 203-204: I do not understand this distinction. The difference from the initial state taken from observations is the model bias?

We modified the sentence as follows "This metric shows to what extent and how quickly the modelled 3D temperature deviates from the ocean initial state as the resolution changes. The anomaly for a specific date is computed as the difference between this current value and the WOA13 temperature."

Line 210: Lellouche et al 2021 ref missing
Reference added

Table 2: is the standard deviation stated the inter annual standard deviation of the global mean SST or the mean standard deviation of the global spatial deviation of SST?

It is the interannual standard deviation of the global mean SST as now specified in the caption "Global annual mean, its standard deviation and linear trend of sea surface temperature for the period 1982-2018 (common to all SST datasets), ..."

Line 254: I don't understand the difference between "model physics", which I generally take to be parameterizations and "unresolved processes" which require parameterization?

Thank you for the comment. The opening sentence of the paragraph was inaccurate and redundant. It was deleted.

Line 260: Most of the SST biases are reduced": This is not visually that obvious. State the rms error of annual mean SST at each resolution.

This is correct. The global averaged SST error is similar among models, GLOB16 is not the lowest. There are improvements at local scales. We added the SST RMSE in the manuscript.

Section 3.3.2 and Figures 5-6: This discussion would be much improved by showing panels with the summer MLD (JJA for NH and JFM for SH) as a single plot and winter MLD (vice versa) as a single plot with the color scale appropriate to each season. The discussion of similarities in summer or lower latitude features is completely obscured by the full annual range color scale. This is also a very long section with little insight beyond that they are different. To what extent can we attribute differences to changes in preconditioning of water masses due to differences in large scale flow versus changes in the restartification power of mixed layer eddies in each case? Does the ORCA1 model include a parameterization (e.g. Fox-Kemper e al) of submesoscale mixed laser restratification included in its GM parameterization?

*Please note that figure numbering has changed in the revised manuscript from Figure 5 onwards; for consistency with the referee's comments, hereafter we use the figure numbers as in the first submitted paper and indicate the new one in brackets.*

Thanks for this comment, which helped to improve the MLD plots. As suggested, we produced a new version of figure 5 [*figure 6 in the revised manuscript*] and figure 6 [*figure 7 in the revised manuscript* that show the winter and summer MLD, respectively, with more appropriate color scale. We decided to keep the March/September spatial distribution of the MLD that is very similar to the JJA/JFM means. The 3-month mean mainly impacts the magnitude of the MLD that is reduced in both observation and models. All plots are attached below for a comparison. The text has been slightly shortened.

A dedicated multi-model study, coordinated by CMCC and based on a larger suite of OMIP simulations at low and high resolutions, will analyze the role that oceanic mesoscale eddies play, in the Labrador Sea, in determining both the location and strength of deep winter convection and the re-stratification of the convected water mass during spring and summer.

The parameterization scheme for sub-mesoscale mixed layer eddies designed by Fox-Kemper et al. (2011) is not included in the CMCC ORCA1 configuration used in this study. This choice is supported by some previous studies based on forced and coupled simulations. The use of the parameterization to improve the mixed layer representation does not consistently result in better performances than models without the parameterization (e.g. Heuzé et al., 2017, Calvert et al.,2020).

- Calvert, D., G. Nurser, M. J. Bell, and B. Fox-Kemper: The impact of a parameterisation of submesoscale mixed layer eddies on mixed layer depths in the NEMO ocean model. *Ocean Modelling*, 154:101678, https://doi.org/10.1016/j.ocemod.2020.101678, 2020.
- Fox-Kemper, B., G. Danabasoglu, R. Ferrari, S. M. Griffies, R. W. Hallberg, M. M. Holland, M. E. Maltrud, S. Peacock, and B. L. Samuels: Parameterization of mixed layer eddies. III: Implementation and impact in global ocean climate simulations. *Ocean Modelling*, 39:61-78, https://doi.org/10.1016/j.ocemod.2010.09.002, 2011.
- Heuzé, C.: North Atlantic deep water formation and AMOC in CMIP5 models. *Ocean Science*, 13, 609–622, https://doi.org/10.5194/os-13-609-2017, 2017.

[Figure]

Figure RC1_1. Map of mean mixed layer depth (in m) for

*A)* March in the Northern Hemisphere and September in the Southern Hemisphere, from (a) observation-based estimates from de Boyer Montégut et al. (2022), (b) GLOB16, (c) ORCA025 and (d) ORCA1. MLD fields are computed as the monthly climatology over last 10-year output.

*B)* as A but averaged over boreal winter (January-February-March) in the Northern Hemisphere and boreal summer (June-July-August) in the Southern Hemisphere.

[Figure]

Figure RC1_2. Map of mean mixed layer depth (in m) for

*A)* September in the Northern Hemisphere and March in the Southern Hemisphere, from (a) observation-based estimates from de Boyer Montégut et al. (2022), (b) GLOB16, (c) ORCA025 and (d) ORCA1. MLD fields are computed as the monthly climatology over last 10-year output.

*B)* as A but averaged over boreal summer (June-July-August) in the Northern Hemisphere and boreal winter (January-February-March) in the Southern Hemisphere.

Line 338-342 discussion of higher order statistics in Johnson and Lyman 2022: What is the relevance to this study none of these statistics are evaluated in the paper

GOSML dataset by Johnson and Lyman (2022) is constructed as a statistical monthly climatology of the global mixed layer depth, temperature, and salinity determined from the Argo profiles using the density algorithm of Holte & Talley (2009). The dataset includes means and variances, plus additional statistics for mixed layer properties including the median (50[th] percentile), 5 [th], and 95 [th] percentiles, as well as skewness and kurtosis. Johnson and Lyman (2022) find that the distribution of MLD is non gaussian, with large skewness and kurtosis that vary seasonally and spatially. The

MLD variance displays seasonal variations and depends on the MLD itself (regions with deep ML have a large MLD variance). The properties of the MLD statistics for the OMIP simulations are analysed in Treguier et al. (2023). Studying the mixed layer statistics is not the aim of this manuscript. Here we used the GOSML only as a reference dataset for validating the latitudinal variability of the simulated mean MLD in March and September, together with the observational dataset compiled by de Boyer Montégut et al. that uses a different mixed layer definition.

- Johnson, G. C., and Lyman, J. M.: GOSML: A global ocean surface mixed layer statistical monthly climatology: Means, percentiles, skewness, and kurtosis. J. Geophys. Res.: Oceans 127. doi: 10.1029/2021JC018219, 2022.

- Treguier, A. M., de Boyer Montégut, C., Bozec, A., Chassignet, E. P., Fox-Kemper, B., McC. Hogg, A., Iovino, D., Kiss, A. E., Le Sommer, J., Li, Y., Lin, P., Lique, C., Liu, H., Serazin, G., Sidorenko, D., Wang, Q., Xu, X., and Yeager, S.: The mixed-layer depth in the Ocean Model Intercomparison Project (OMIP): impact of resolving mesoscale eddies, Geosci. Model Dev., 16, 3849–3872, https://doi.org/10.5194/gmd-16-3849-2023, 2023.

Line 349: Johnson and Lyman 2022 ref missing
Reference added

Line 350: again, the use of the full dynamic range in the axis scale makes it difficult to compare the quality of the simulation of shallower mixed layers. The relative error could be just as large as for deeper M.
We modified the MLD plots with color scales that are now appropriate to each season. Please see above the comment on Figure 5 [*figure 6 in the revised manuscript*] and Figure 6 [*figure 7 in the revised manuscript*].

Line 374 "representations is underrepresented": nonsensical phrase
We reworded the sentence.

Line 377: "unable to represent flow instabilities …": yes, but the figure is showing mean low speed, not EKE
In accordance with the Referee's suggestion, we changed the sentence as follows "It captures the major current systems of the global ocean, but it underestimates the magnitude of the surface velocity field and fails to represent mesoscale eddies and meanders."

Line 385: "dependent on model numerics": how so? Don't all of the models use the same numerical methods to solve the equations of motion?
Among the three configurations, the numerical methods are similar but not exactly the same. The western boundary currents differences are also dependent, for example, on differences in the lateral boundary conditions and topography. More details on the physical parameters are now provided in Table 1.

Line 386: "impact of mesoscale dynamics": This has not been shown. It could simply be the impact of viscous boundary layer dynamics or topography which also differ across configurations
Following the comment, we reworded as "The Gulf Stream simulated by the three models is depicted in Figure 10 (left column)."

Line 404 "passed": past
Done

Line 421 "Figure 10 shows role of mesoscale eddy field …": again, the figure shows the mean flow and no analysis of eddy-mean flow interaction is provided.
Thank you for the comment. We reworded as "Figure 11 shows the complex ocean circulation in the Southern Ocean sector...".

Section 3.3.2 (sigma overturning): A more precise definition of how the stream function was calculated at each resolution is required. Was it computed from Eulerian mean (monthly, annual, climatological?) velocity and density fields? Was the GM eddy-induced velocity included in the ORCA1 result? The authors should write down the integral with averaging operators in the appropriate places for clarity. This is important in trying to understand whether differences seen are related to "bolus velocity", diapycnal processes or surface forcing. The reference to Andrews and McIntyre suggest that we should be interpreting something about eddy induced transport, but the discussion is unclear. One of the major differences is the structure of the strong clockwise cell in the ACC region which I presume is related to the degree of compensation of the Deacon cell. No discussion of this feature is provided, nor are we sure how to interpret the result given the uncertainty about exactly what is being shown.
The MOC in density space is computed following the formulation by Farneti et al 2015 (equation 7 in Appendix A). In section 3.3.2, we added this reference to the computation of the MOC on potential density surfaces (referenced to 2000 dbar) from monthly meridional velocity and density fields.
We also specified that the meridional velocity in output to the ORCA1 model is the sum of the Eulerian-mean velocity and the GM eddy-induced component obtained through GM parameterization (Gent and McWilliams 1990). The reference to Andrews and McIntyre (1978) was wrongly placed.
The use of potential density as the vertical coordinate, rather than depth, results in a better characterization of water mass transport and is more suitable for representing the MOC in the Southern Ocean. In particular, the wind-driven Deacon cell, which normally appears in depth-space MOC, is mostly due to a geometrical effect of the east-west slope of the isopycnals when the zonal and vertical integration is computed at fixed depth levels. No cross-isopycnal flow is associated with it (Döös and Webb 1994, Farneti et al. 2015).
The MOC structure presented in Figure 11 [*figure 12 in the revised manuscript*], from the southernmost boundary to 30ºS, agrees with previous studies (e.g. Farneti et al. 2015). The wind-driven subtropical cell is part of the horizontal subtropical gyres and is confined to the lightest density classes. This anticlockwise cell comprises a surface flow spreading poleward to 40S, compensated by an equatorward return flow. Below, the upper cell is depicted by the large clockwise circulation, which mainly consists of upper circumpolar deep water. The anticlockwise lower cell, in the densest layers, that consists of the poleward lower circumpolar deep water and the deeper equatorward AABW. From 60ºS to the Antarctic continent, the transport represents the contribution of subpolar gyres in the Weddell and Ross Seas. The revised text includes this description and a better characterization of the water masses.
The global MOC for the three simulations is presented in the plots below in depth space (left) and potential density space (right). In all models, south of 30S, the MOC in depth space shows the clockwise Deacon and upper cells, and the anti-clockwise lower, subtropical and subpolar cells. In density-space, the Deacon Cell disappears, and the surface waters recirculates in the ACC anticlockwise upper cell. This is in very good agreement with the schematics of the main Southern Ocean cells (see Figure 16 in Farneti et al. 2015).

[Figure]

Figure RC1_3. Time-mean zonally integrated MOC computed in depth space (left) and in density space as function of $\sigma_2$ for the global ocean as reproduced by the three simulations. Overturning is averaged over the last 10 years of simulation (2009–2018).

- Farneti, R., Downes, S. M., Griffies, S. M., Marsland, S. J., et al..: An assessment of Antarctic Circumpolar Current and Southern Ocean meridional overturning circulation during 1988–2007 in a suite of interannual CORE-1125 II simulations, Ocean Model., 93, 84–120, https://doi.org/10.1016/j.ocemod.2015.07.009, 2015.

Line 466: "This suggests that longer integrations are required for GLOB16 …": It was previously stated that the analysis of all cases was for the first cycle (Line 150). Why is that only GLOB16 requires longer integration to be compared?
We followed this comment and rephrased the imprecise sentence as follows "While the upper ocean takes decades to achieve equilibrium, the deep ocean adjustment requires hundreds of years to reach a quasi-equilibrium state (e.g. Danabasoglu et al., 1996) because of the slow diffusion of active tracers. Tsujino et al. (2020) show that OMIP2 low-resolution simulations take about four cycles to spin-up, and the AMOC declines in the first cycle and slowly recovers thereafter. A longer GLOB16 integration would be necessary to reach a quasi-equilibrium behavior of the overturning in the deep ocean and analyze the long-term evolution of deep-water properties from the initial state also in the eddying ocean."

- Danabasoglu, G., J. C. McWilliams, and W. G. Large: Approach to equilibrium in accelerated global oceanic models. J. Climate, 9, 1092–1110, https://doi.org/10.1175/1520-0442(1996)009<1092:ATEIAG>2.0.CO;2, 1996.

- Tsujino, H., Urakawa, L. S., Griffies, S. M., Danabasoglu, G., et al.: Evaluation of global ocean–sea-ice model simulations based on the experimental protocols of the Ocean Model Intercomparison Project phase 2 (OMIP-2), Geosci. Model Dev., 13, 3643–3708, https://doi.org/10.5194/gmd-13-3643-2020, 2020.

Figures 13-14 and accompanying discussion: This is a long discussion of the GLOB16 results alone with no explicit comparison across resolutions. It seems unnecessary if the purpose of the paper is investigating resolution dependence rather than an assessment of GLOB16 alone.

The main purposes of this paper are to evaluate the GLOB16 model performance and to document if and how the CMCC "eddy-resolving" ocean model resolution change the representation of large-scale ocean variability with respect to observations and lower-resolution models, highlighting the relative advantages and disadvantages of running ocean–sea ice models at such resolution.

In agreement with previous studies, our results show that a number of prominent biases and model errors persist, or even worsen, despite increases in model resolution. However, the finer resolution remains one possible way in which model capabilities can be enhanced, thanks to the explicit representation of eddies.

In Figure 13 [*figure 14 in the revised manuscript*] and Figure 14b [*figure 15b in the revised manuscript*], we present the time evolution of the Atlantic MOC and MHT at fixed latitudes for GLOB16 in comparison to available estimates. We decided to not show the ORCA1 and ORCA025 results because the interannual variability of these two metrics is not largely affected by the resolution (see figures below).

Mean values of both quantities at 26.5N are included in the text for observation and all models.

[Figure]

Figure RC1_4. time evolution of monthly mean AMOC transports, defined as the maximum value of the global overturning stream function in GLOB16 (blue line) ORCA025 (green line) and ORCA1 (orange line) computed (a) across 26.5°N and compared to RAPID estimates, and (b) 34°S compared to SAMBA record.

[Figure]

Figure RC1_5. Times series of the monthly-mean total AMHT in GLOB16 (blue line), ORCA025 (green line) and ORCA1 (orange line) across 26.5°, compared to the RAPID record (magenta).

Line 560: Treguier 2012 and Robert 2016 refs missing
References added

Dear Editor and Reviewer,

We would like to thank you for accurately reading and commenting the manuscript, and suggesting how to improve it. Answers to your comments are given in details hereafter. We hope that you will find them satisfactory. All authors agree with the modifications made to the manuscript. Reviewer comments are in black and are followed by our response (in blue) that includes changes and/or additions to the text.

For the authors,
Doroteaciro Iovino

**RC2**: 'Comment on egusphere-2023-469', Anonymous Referee #2, 19 May 2023

This manuscript provides a high-level assessment of the performance of the CMCC GLOB16 1/16° global ocean and sea ice simulation relative to two related simulations at 1° and 0.25° (ORCA1 and ORCA025) and range of observationally estimated values. The text is clearly written and logically organised, and the figures are appropriate and well-presented. While the manuscript provides few new insights, it gives a good overview of the main strengths and weaknesses of GLOB16 which will be of use to users of that model and as a point of comparison for other modelling efforts internationally. I would like to see it published after major revision to include more details of interest to the ocean modelling community, and I've provided detailed suggestions in the hope that this will help.

While the comparisons to observation are well-chosen and cover many of the most important assessment criteria for models of this sort, there are some significant omissions:

- Several references are made to sea ice processes and differences between the two sea ice models as explanations for differing ocean features, but no sea ice results are presented to back up these speculations. Comparing sea ice results (such as climatological maps of concentration and time series of ice volume, area and extent) with observations and between LIM2 and CICE would be very helpful to aid interpretation of the ocean biases and understand the differences between these models.

We agree with the referee. Understanding of sea ice-ocean interactions in both polar regions is an important step to improve our knowledge of the climate processes at regional and global scales. Changes in sea ice can greatly affect ocean hydrography, ocean dynamics and heat transport. Nevertheless, the sea ice models used in this study have large differences in their complexity (melt pond, ice thickness distribution, salt bulk, etc.) and they were set with their default initial conditions that do not correspond for sea ice thickness. Of course, all these differences together affect the connection and interplay between sea ice and atmospheric/oceanic drivers. A detailed comparison of the sea ice model complexity and impact on the ocean is behind the scope of the paper – the impact of the model complexity and model differences has been addressed in several other studies (e.g. Uotila et al. 2017, Massonnet et al. 2011). Only few ice variables were saved from the OMIP simulations, and a detailed comparison of sea ice thermodynamics and ocean-ice fluxes is not possible. Nevertheless, following this suggestion, we include plots of the time evolution of sea ice extent and its 2D spatial distribution.

- Uotila, P., Iovino, D., Vancoppenolle, M., Lensu, M., and Rousset, C.: Comparing sea ice, hydrography and circulation between NEMO3.6 LIM3 and LIM2, Geosci. Model Dev., 10, 1009–1031, https://doi.org/10.5194/gmd-10-1009-2017, 2017.
- Massonnet, F., Fichefet, T., Goosse, H., Vancoppenolle, M., Mathiot, P., and König Beatty, C.: On the influence of model physics on simulations of Arctic and Antarctic sea ice, The Cryosphere, 5, 687–699, https://doi.org/10.5194/tc-5-687-2011, 2011.

- There could also be more comparison of the variability at different resolutions relative to observations, e.g. maps of sea level standard deviation, to highlight the impact of increased resolution.

We agree with the referee that the manuscript could benefit from a more comprehensive comparison. We included new metrics trying not to alter the manuscript structure too much or to increase its length. A new paragraph on sea ice properties is included where satellite observations are taken into account. We also computed the standard deviation of the sea surface height, as

suggested, for the three models and AVISO (http://www.aviso.altimetry.fr/). The new plots (Figure RC2_1 here and Figure 5 in the revised manuscript **)** are shown below.

The STD of the daily SSH shows the amplitudes of mesoscale eddies from AVISO and models. This variability is associated with high kinetic energy. GLOB16 shows a significant improvement in the position, strength, and variability of the western boundary currents, the Antarctic Circumpolar Current and the Zapiola gyre. The SSH STD is close in GLOB16 to what can be observed from altimetry, while it decreases substantially in the coarser-resolution experiments. This figure is now included and discussed in Section 3.2.

[Figure]

**Figure RC2_1** Standard deviation of daily SSH (in m) from AVISO and the three model configurations during 2009-2018.

- No information on model computational performance is provided, e.g. the relative core-hours per simulated year at each resolution. This would be useful for practitioners in deciding whether the improved solution fidelity at high resolution is worth trading off against a shorter simulation duration within finite computational and walltime budgets. Other technical details (e.g. number of computational cores, parallelisation efficiency, etc) would also be helpful.

All simulations were performed on the CMCC Zeus HPC platform, equipped with two Intel Xeon Gold 6154 (3.0 GHz, 18 cores) and 96 GB of main memory per node. The interconnection network is the 100Gbps Infiniband EDR, while the file system is the IBM General Parallel File System (GPFS). The models were compiled with the Intel compiler suite version 20.1 and MPI library (based on MPI version 3.1).

The table RC2_1 reports the computational performance of the three configurations for the OMIP2 production runs. These values are now included and discussed in the manuscript.

Given the limited computational resources available and the need to divide these resources among the three simulations, these are not fully representative of the best achievable performances. It is worth mentioning that the computational performance of a coupled modeling system as the one used for ORCA1 and ORCA025 depends on the performances of any single model component and the efficiency of the coupling software. The NEMO and CICE codes run sequentially. In the lowest-resolution, NEMO uses ~78% and CICE ~7.5% of the wall time; the remaining is given to the atmospheric and river data models and to the coupler. In ORCA025, the ocean and ice models use 72% and 15.3%, respectively. In the GLOB16 framework, LIM is not a stand-alone model, it is a

module of the ocean code. The two components are interactively interfaced without using a coupling code; LIM2 takes almost 20% of the wall time.
We included this technical information in a dedicated paragraph "Computational performance" in section 2.

**Table RC2_1.** Performance for the three production models. For ORCA1 and ORCA025, the number of cores used by the ice model is indicated in parenthesis.

| Configuration | Number of cores | Wall time (h y$^{-1}$) |
|---|---|---|
| ORCA1 | 128 (96) | 1.31 |
| ORCA025 | 1008 (972) | 4.44 |
| GLOB16 | 2086 | 94.22 |

The depth of analysis could also be improved. Many parts of the text simply give a verbal description of what the plots show. This does not provide much value to the reader, and in some places the descriptions are also incorrect. Additional physical insights and interpretations would be very helpful.
We paid more attention to the text, corrected the description of the model output when needed, and we better articulated the interpretations of model results in section 3.

An assessment of the impact of increased resolution in GLOB16 is hampered by many important differences between this model and ORCA1 and ORCA025 (e.g. a different sea ice model, twice as many vertical levels, many different parameterisations). This is unfortunate but perhaps beyond the scope of this paper to rectify.
The different configurations have not been developed simultaneously and with no similar scientific purpose. The model system used for our OMIP runs at low and medium resolution is based on the CMCC coupled system used in the CMIP6 exercises (Eyring et al. 2016). The OMIP high-resolution was informally organized by the CLIVAR Ocean Model Development Panel (https://www.clivar.org/clivar-panels/omdp), with no well-defined set-up and spin-up protocols apart from the use of JRA55-do forcing. By the time we started the OMIP2 simulations, including GLOB16 code in the framework of the coupled system was not affordable for us. We run the tested and proven GLOB16 ocean configuration with its default NEMO-LIM framework, as used for the short-term ocean forecast and research objectives.
The differences in the model implementation and set-up impact the results and limit the model intercomparison, but we believe that this model study can still provide insight in the relative benefits and drawbacks of running ocean–sea ice models at eddy-rich resolution, and that the metrics used in the paper are robust enough to highlight the impact of grid refinements, even if not to isolate it.

- Eyring, V., Bony, S., Meehl, G. A., Senior, C. A., Stevens, B., Stouffer, R. J., and Taylor, K. E.: Overview of the Coupled Model Intercomparison Project Phase 6 (CMIP6) experimental design and organization, Geosci. Model Dev., 9, 1937–1958, https://doi.org/10.5194/gmd-9-1937-2016, 2016.

Re. Code and Data Availability - can the configuration parameter files for GLOB16, ORCA1 and ORCA025 be provided? And links to source code for LIM2 and CICE4?
We added the links to the LIM2 and CICE4 source codes in Code and Data Availability. We update table 1 with more configuration parameters.

There are many small typos and errors, which I have listed in the last section below.
Numbers in my detailed comments below are line numbers.

Comments on content:
56: Mention that this multi-resolution approach is similar to that taken at other modelling centers, eg Storkey et al 2018 https://doi.org/10.5194/gmd-11-3187-2018, Adcroft et al 2019 http://dx.doi.org/10.1029/2019MS001726, Kiss et al 2020 https://www.geosci-model-dev.net/13/401/2020/

Yes, the multi-resolution approach is common to many modeling centers, GFDL (Adcroft et al. 2019) and UK MetOffice (Storkey et al. 2018) among them. In addition to the resolution comparison, the first two papers are intended to document model developments, updates and new implementations, while Kiss et al. (2020), as Li et al. (2020), both mentioned in our manuscript, focus on the description of OMIP simulations. We rephrase the sentence as follows "Under this framework, several modeling centers started to perform multi-resolution studies (e.g., Storkey et al. 2018, Adcroft et al. 2019, Kiss et al. 2020, Li et al. 2020). Following the same approach, CMCC uses a hierarchy of ocean-sea ice configurations..."

- Li, Y. W., and Coauthors, 2020: Eddy-resolving simulation of CAS-LICOM3 for phase 2 of the Ocean Model Intercomparison Project. Adv. Atmos. Sci., 37(10), 1067–1080, https://doi.org/10.1007/s00376-020-0057-z

84: if the model has been "extensively upgraded", then Iovino et al. 2016 is not very useful as a reference. Please provide a list of the important changes that have been made since Iovino et al. 2016

The sentence was unclear. The idea was to mention that the configuration is now running with the most recent NEMO code, but this is not relevant for this study. We rephrased as follows "The model is based on its first implementation documented in Iovino et al. (2016), where the ocean component is upgraded from version 3.4 to version 3.6-stable"

93: give the equation of state, and say what the prognostic variables are (e.g. conservative or potential temperature)

We added in the model description that we use the EOS80 equation of state and the prognostic state variables are potential temperature and practical salinity.

105: "While the best approach to identify the impact of grid resolution should be to change only resolution and associated physics in the suit of models, this was not the case in similar previous studies (Chassignet et al., 2020, Kiss et al., 2020, Li et al., 2020)." - while there were unavoidable differences in parameterisation due to the differing ability to resolve processes at different resolutions, Kiss et al. 2020 made significant efforts to harmonise the configurations across resolution. It sounds like an effort in that direction would be possible here and would more cleanly highlight the effects of improving resolution. Is that beyond the scope of this project?

We do agree that this intercomparison study would have benefited from a hierarchy of identical or very similar ocean-ice configurations, with the only differences being the grid resolution and related parameters. As said above, introducing and testing GLOB16 into the coupled system was not feasible. Based on the shortcomings highlighted by this study, a proper harmonization of the ocean-ice configurations might be considered in the CMCC strategy for the next OMIP exercises.

Table 1: as also pointed out by Alexander Shchepetkin, "barotropic sub-step [sec]" should be "barotropic sub-steps". It would also be worth noting that this was a configuration error (the barotropic timestep is an order of magnitude smaller than needed for CFL stability) which would adversely affect the computational cost, but not the numerical solutions.

The barotropic time steps used in the ORCA025 and GLOB16 runs are smaller than those needed to satisfy the CFL stability condition. The choice of the ratio baroclinic/barotropic time steps was based on larger time steps that we have not been able to achieve at run time for the OMIP2 simulations. As stated by the reviewer, this choice impacts the computational performance, but not the numerical results. This information does not provide useful indications and suggestions for future runs, hence we removed it from the table in the revised version.

186: does the differing surface vertical resolution play a role in this SST difference?

While several studies have documented the impact of the horizontal oceanic resolution on the oceanic thermal structure and 3D circulation, only few studies have assessed the impact of the vertical resolution (e.g. Jia et al. 2021, Chassignet and Xu 2021, Stewart and Hogg 2019).

Stewart et al. (2017) pointed out that the effort to increase the horizontal resolution can be undermined if unaccompanied by adequate refinements to the vertical resolution. The change in vertical grid spacing can significantly impact the stratification and vertical shear. Then, the vertical resolution near the ocean surface is one of the major factors that impact the SST simulation in numerical models.

In the coarser resolution models, the choice of the vertical grid is mainly based on the computational cost required to run the complete set of CMIP exercises. The 50 vertical levels in ORCA1 and ORCA025 are the minimum requirement by Stewart et al. (2017) to resolve the first baroclinic mode in a z-coordinate model. In GLOB16, the vertical grid was constructed to adequately resolve the upper ocean dynamics and vertical structure of the ocean currents in short-term simulations.

The impact of vertical resolution on the surface and subsurface temperatures was not tested in any of our configurations. Due to computational resources, we did not perform specific simulations to isolate the effects of horizontal resolution from the vertical one and vice versa.

One may expect that the 98 vertical levels in GLOB16 can better capture the detailed structure of vertical temperature profiles, resolve the fine details of the stratification in the upper ocean, contributing to the changes in the vertical mixing structure.

- Chassignet, E.P., and X. Xu. On the Importance of High-Resolution in Large-Scale Ocean Models. Adv. Atmos. Sci. 38, 1621–1634. https://doi.org/10.1007/s00376-021-0385-7, 2021.
- Jia, Y., Richard, K., and H. Annamalai. The Impact of Vertical Resolution in Reducing Biases in Sea Surface Temperature in a Tropical Pacific Ocean Model. Ocean Modelling, 157, doi: 10.1016/j.ocemodel.2020.101722, 2021
- Stewart, K. D., and A. McC Hogg. Southern Ocean heat and momentum uptake are sensitive to the vertical resolution at the ocean surface. Ocean Model. 143, 101456, 2019.
- Stewart, K. D., Hogg, A. McC., Griffies, S. M., Heerdegen, A. P., Ward, M. L., Spence, P., and England, M. H.: Vertical resolution of baroclinic modes in global ocean models, Ocean Model., 113, 50–65, https://doi.org/10.1016/j.ocemod.2017.03.012, 2017.

190: "Kiss et al. (2020) where the 1/10° ocean surface ... with the largest bias from observations" no, obs in Kiss et al 2020 fig 3b is an anomaly offset by 18°, so this plot only compares trend, not bias.

Thank you for the comment. We modified the sentence.

157: specify whether the salt restoring is constrained to have zero total flux. If not, how significant is the salt flux for the drift in total ocean salt mass?
The sentence "Salinity restoring is applied globally (excluding sea ice covered areas) via a salt flux" was modified in "Salinity restoring is applied globally via an equivalent surface freshwater flux. There is no salinity restoring below sea ice". Although the salt restoring term is not constrained to have zero net flux, the drift in the volume averaged global salinity at the end of the first forcing cycle is not significant, amounting to is $6.04e^{-3}$ psu in ORCA1, $-3.81e^{-6}$ psu in ORCA025 and $-2.94e^{-4}$ psu in GLOB16.

228: "The SSS drift is offset by the surface salinity restoring that is incorporated into the codes to enforce salt conservation in the model ocean (in Sect. 2)." - do you mean "constrain SSS drift" rather than "enforce salt conservation"? Conserving total ocean salt mass would require balancing any net flux into sea ice and nonzero net restoring salt flux.
Following the comment, we changed the sentence as follows "The SSS drift is offset by the surface salinity restoring that is incorporated into the codes to constrain the salinity drift in the model ocean".

229: "The restoring of SSS drives its quasi-stationary evolution, the salt exchange between ocean and sea-ice due to ice formation and melting, is the only source of salt for the ocean." - so is there zero net salt restoring flux?
The salt restoring flux is not zero, but the ocean-sea ice salt exchanges are the most relevant source of salt to the ocean.

254: drift may also be due to biased forcing (as mentioned later in this paragraph), or adjustment from an unbalanced initial condition
The opening sentence of the paragraph was deleted during the review process.

257: or initial condition... though less likely for surface bias
Corrected.

Fig 4: if this bias was plotted relative to the WOA initial condition it would remove any effect from initial condition bias relative to ERSSTv5
We do agree that the model bias computed against the WOA sea surface temperature removes the effect from the initial condition (IC) bias relative to the observational dataset. Figure RC2_2 (b-d) shows the longitude-latitude maps of the differences of the simulated SST and the WOA SST over the period 2009-2018. While the large-scale spatial patterns of temperature biases are similar to the ones relative to ERSSTv5 (Figure 4 in the manuscript), the bias amplitude confirms that the ocean surface temperature increases from the beginning to the end of the integration period, as shown in Figure 1b. GLOB16 exhibits the largest increase of temperature at the low and middle latitudes, but also a clear improvement along the Gulf Stream and Kuroshio extension and in the ACC region, and in the North Atlantic subpolar gyre. In addition to Figure 1b and Figure 2, we prefer to

evaluate the ability of the models to reproduce the ocean surface characteristics in comparison to ERSSTv5 as an *independent* data set.

[Figure]

Figure RC2_2. Climatological annual mean SST (in °C) from WOA13 (a) and the biases of the simulated surface temperature (b-d) against WOA13, computed over the 2009–2018 time period.

Figs 5,6: Summer differences are very hard to discern. It would be better to plot winter in both hemispheres in fig 5 and summer in both hemispheres in fig 6, with different color scale.

*Please note that figure numbering has changed in the revised manuscript from Figure 5 onwards; for consistency with the referee's comments, hereafter we use the figure numbers as in the first submitted paper and indicate the new one in brackets.*

As suggested by the referee, the new figure 5 [*figure 6 in the revised manuscript*] shows the mixed layer structure in winter (March in the northern hemisphere and September in the southern hemisphere). The new figure 6 [*figure 7 in the revised manuscript*] shows the mixed layer structure in summer (September in the northern hemisphere and March in the southern hemisphere). The two MLD plots now have two different palettes. New Figure 5 [*figure 6 in the revised manuscript*] and Figure 6 [*figure 7 in the revised manuscript*] follow.

A)

[Figure]

Figure 5. Mean Mixed layer depth (in m) averaged over March (in the Northern Hemisphere) and September (in the Southern Hemisphere) from (a) the de Boyer Montégut et al. (2022) climatology, (b) GLOB16, (c) ORCA025 and (d) ORCA1. MLD fields are computed as the monthly climatology over last 10-year output.

B)

[Figure]

Figure 6. Mean mixed layer depth (in m) averaged over September (in the Northern Hemisphere) and March (in the Southern Hemisphere) from (a) the de Boyer Montégut et al. (2022) climatology, (b) GLOB16, (c) ORCA025 and (d) ORCA1. MLD fields are computed as the monthly climatology over last 10-year output.

Figs 5,6: extend latitude range further south to show Ross and Weddell convection
Done, the new plots include all the Antarctic coast line, including the ice sheet line in the Weddell Sea.

Fig 5,6: add citation for (a) to caption
Done

Fig 7 adds little beyond what is in 5&6, other than an additional obs dataset - could be supplementary material.

Thank you for the suggestion. We prefer to keep Figure 7 [*figure 8 in the revised manuscript*] in the manuscript without having a supplementary material section mainly to evaluate the latitudinal variation of the mixed layer against a new monthly mixed layer climatology. The Global Ocean Surface Mixed Layer Statistical Monthly Climatology (Johnson and Lyman. 2022) is still based on Argo CTD data, but the mixed layer properties are computed using the density algorithm from the hybrid method by Holte and Talley (2009). For the winter mixed layer, the differences between the two datasets used in Figure 7 are as large as the spread among model configurations at middle and high latitudes.

- Holte, J., and Talley, L. (2009). A New Algorithm for Finding Mixed Layer Depths with 553 Applications to Argo Data and Subantarctic Mode Water Formation. Journal of 554 Atmospheric and Oceanic Technology, 26(9), 1920–1939. 555 https://doi.org/10.1175/2009jtecho543.1
- Johnson, G. C. and J. M. Lyman. 2022. GOSML: A Global Ocean Surface Mixed Layer Statistical Monthly Climatology: Means, Percentiles, Skewness, and Kurtosis. Journal of Geophysical Research, 127, e2021JC018219. https://doi.org/10.1029/2021JC018219.

Fig 7: cite obs datasets in caption

Done

279: "The observed MLD is diagnosed through a density threshold criterion as the depth over which the potential density increases by 0.03 kg m-3 from the reference value of surface potential density taken at 10m depth; resulting values are mapped on a monthly basis at 1°x1° spatial resolution (de Boyer Montégut et al., 2004). The same density threshold method is applied to model output as recommended by Griffies et al. (2016) to compute the MLD in OMIP models." - This is not the same method because Griffies et al 2016 specify using the top model level as a reference, not 10m. Treguier et al 2023 state CMCC use the top model level, not 10m. Or was MLD recalculated to use 10m (ie different from the Griffies et al method)? If not, this difference may contribute to the bias, as Treguier et al discuss. Please clarify what was done and its bearing on comparison with obs.

All three NEMO configurations compute the depth of the mixed layer following a set of criteria. GLOB16 output the MLD based on the $0.03\,\mathrm{kg\,m-3}$ density threshold relative to a depth of $10\,\mathrm{m}$ and to the top model level (0.5m) for all the integration. ORCA1 and ORCA025 provide the MLD based on the same criteria, but the MLD computed with 10m reference depth is saved only in the last 10 years of simulation, with daily output.

Griffies et al (2016) suggested to use the top level as a reference in the OMIP models. We do not follow this suggestion and so we removed the reference. In this manuscript, we decided to use the 10m reference depth for all simulations to be consistent with the methodology used by de Boyer Montegut (2022).

The comparison between the MLD computed with the same density threshold by two different reference depths is presented in Treguier et al. (2023) for the CMCC low-resolution configuration, where the MLD was computed offline for both criteria. Below, Figure RC2_3 and Figure RC2_4 show the differences between the two criteria applied to all CMCC configurations.

[Figure]

[Figure]

Figure RC2_3. Climatology of the observed MLD (a) and difference (in m) between the MLD computed in the three configurations using the density threshold with two reference depths: 10 m and the top model level, in March in the northern hemisphere and September in the southern hemisphere.

[Figure]

Figure RC2_4. as figure RC2_3 but for September in the northern hemisphere and March in the southern hemisphere.

297: also mention MLD bias at 0°E in mid-north Greenland Sea: several hundred metres too shallow

The sentence was modified as follows "...with the closest agreement in the Irminger Sea and a negative bias in the Greenland Sea".

367: could also comment on the differences in the Zapiola anticyclone at different resolution - seem similar to what was seen by Kiss et al 2020.

We added the Zapiola anticyclone in this sentence. It was also mentioned (line 434 in the first submitted manuscript) as "Downstream of Drake Passage, the ACC northern part describes an equatorward loop to ~40°S in the southwestern Argentine Basin". This sentence was deleted and replaced by "Downstream of Drake Passage, GLOB16 accurately reproduces the ACC northern branch that breaks off as the Malvinas current and flows northward along the edge of the Patagonian shelf. In the southwestern Argentine Basin, the eddy-driven Zapiola anticyclone is well-placed between the 40°-50°S and its spatial structure and strength are in close agreement with the OSCAR field."

384: "In an eddy-rich regime, the ocean model is less diffusive/viscous" is back to front: At high res, diffusivity/viscosity can be reduced while maintaining numerical stability; this allows WBCs to be realistically narrow and inertial, and the resolution of the internal Rossby radius allows baroclinic instability and an eddying flow

The initial sentence of this paragraph was rewritten as follows "It is widely recognized that the horizontal grid spacing, sufficient to resolve the Rossby radius of deformation in most of the global domain and allow for a proper representation of baroclinic instability, results in a significant improvement in western boundary currents and associated eddies (e.g. Hurlburt and Hogan, 2000; Yu et al., 2012; Chassignet and Xu, 2017). A proper representation of the WBCs in global ocean models is the result of many contributing factors. Despite the general improvements in their representation due to model resolution, the simulated WBCs strength, width, position and separation remain dependent on a variety of parameter choices made in the numerical models (e.g. Bryan et al. 2007; Chassignet and Marshall 2008), such as boundary conditions, coastline and bottom geometry, friction parametrization, etc. Accurately simulating the Gulf Stream separation in ocean numerical models has been a challenge and does still remain an issue despite the fact that major improvements are realized in eddy-rich ocean configurations (e.g., Chassignet and Xu, 2021). The Gulf Stream as simulated by the three models is presented in Figure 10 (left column)."

384: "less diffusive/viscous leading to an improvement in the strength and position of WBCs" - could relate this to narrowing the Munk (Laplacian) or Haidvogel et al 1992 (biharmonic) viscous WBC scales (A_lap/beta)^(1/3) and (A_bih/beta)^(1/5). Perhaps these scale values could also be useful in table 1:

| | | |
|---|---|---|
| ORCA1: | Munk: | (1e4/2e-11)^1/3~80km |
| ORCA025: | Haidvogel: | (1.8e11/2e-11)^1/5~25km |
| GLOB16: | Haidvogel: | (0.5e10/2e-11)^1/5~12km |

Haidvogel et al 1992 ref:
http://dx.doi.org/10.1175/1520-0485(1992)022%3C0882:BCSIAQ%3E2.0.CO;2

Thank you for the suggestion. The revised version of Table 1 includes the horizontal viscosity values from which the Munk/Haidvogel scales can be derived. In our time-averaged analysis, the narrowing of the WBCs is not the most evident improvement (e.g. Figure 9 [*figure 10 in the revised manuscript*]), it is mentioned for the Gulf Stream at line 551 (submitted manuscript). However, we rewrote the initial sentences of this paragraph (see previous comment).

434: "The Antarctic coastal current is also clearly represented, it flows westward along the Antarctic coast and meets the eastward-flowing ACC at the Drake Passage, emerging as the Malvinas current." - isn't the Malvinas fed by the Antarctic circumpolar current, not the Antarctic coastal current?

We apologize for this oversight. We corrected the sentence removing "emerging as the Malvinas current".

465: More discussion of the overturning south of 30°S would be good, e.g. in comparison with Farneti et al 2015. What depth does 1036.8kg/m^3 correspond to in GLOB16 in this region? Why is this subpolar cell mostly disconnected from the abyssal export cell in GLOB16 and ORCA1? It seems more connected to the surface cell at ~50S, 1036kg/m^3 than the abyss. The circulation below 1037kg/m^3 looks quite different from Kiss et al 2020 fig 7 and many of the models in Farneti et al 2015 fig 17 (and ORCA1 looks very different from the 1° CMCC model Farneti et al show) - can you point this out and speculate as to why? Also in ORCA1 a much larger fraction of the southern ocean upwelling is recirculated south of 35S in an intense clockwise cell rather than joining the NADW cell, and this overturning cell is stronger than at higher resolution - why?

The subpolar and abyssal cells can appear disconnected in the MOC in density space when the upper cell is intense and deep enough to include water that is as dense as the ones formed in the Antarctic region. The global MOC in depth-space is shown in Figure RC2_5 to clearly present the structure of the clockwise Deacon and Upper cells and anti-clockwise subpolar and abyssal cells.

The CMCC simulation in Farneti et al. (2015) was performed within the COREII framework, using a previous version of the ocean-sea ice codes, and the coupling code. Furthermore, Figure 17 in Farneti et al (2015) shows the overturning circulation computed as time mean between 1958-2007, in the fifth COREII cycle integration. Anyway, we think that there are not large differences with the current MOC in the Southern Ocean, beside the vanishing abyssal cell between 60° and 50°S in the newest simulation. This difference may be due to the representation of coastal and near-shore processes, mixing, sea-ice (thermos)dynamics. It is difficult to relate it to a single process or model modification.

In Kiss et al (2020), the circulation denser than 1037 kg m$^{-3}$ is mostly related to the subpolar cell, while the abyssal cell (slightly lighter) is poorly connected with lower latitudes with a transport of dense water of ~6 Sv across 30°S.

We added a more detailed discussion on the overturning in the Southern Ocean with the attempt to reply to the questions addressed here.

[Figure]

Figure RC2_5 Time-mean zonally-integrated overturning circulation (in Sv) as a function of latitude and depth averaged over the period 2009-2018 in the global domain.

589: is it feasible to do a RAPID-like calculation with your model data to verify this?
The calculation is computational demanding and requires retrieving 3D fields of temperature, salinity, meridional and zonal velocities from the archive. The GLOB16 model configuration used for the OMIP simulation is similar enough to the one used in Stepanov et al. (2016) to assume that the RAPID-like calculation will provide similar results. This might be confirmed in the future based on new simulations currently running with updated versions of ORCAx and GLOB16.

- Stepanov, V. N., D. Iovino, S. Masina, et al.: Methods of calculation of the Atlantic meridional heat and volume transports from ocean models at 26.5°N. J. Geophys. Res. Oceans **121**, 1459. https://doi.org/10.1002/2015JC011007, 2016.

Further details needed:
86: specify the latitude at which the tripolar cap starts
Done (see answer to next comment)

87: is this a Mercator grid? If so, say so. e.g. replace "increases poleward as cosine" -> "meridional spacing decreases poleward as the cosine"
The grid description was modified as follows "GLOB16 makes use of a nonuniform tripolar grid with a nominal 1/16° horizontal resolution (6.9 km at the equator reducing poleward). The grid consists of an isotropic Mercator grid between 60°S and 20°N (the meridional scale factor is fixed at 3 km south of 60°S), and a non-geographic quasi-isotropic grid north of 20°N."

105, 121: state the ice grid used - is it the same as the ocean model? If not, what interpolation methods are used?

The LIM2 grid uses the same ocean C-Arakawa grid, while CICE uses a B-Arakawa grid that shares the tracer points with the ocean.

105, 121: state what fields are coupled between the ocean and sea ice in each direction

The coupling interface between NEMO and CICE is described in Cherchi et al. (2019) and references therein (Fogli and Iovino, 2014). The fields exchanged from NEMO to CICE are the sea surface horizontal currents, temperature and salinity, the horizontal components of the gradient of the SSH and the freezing/melting potential heat flux. The fields sent from CICE to NEMO are the momentum, heat and freshwater fluxes from the atmosphere once the evolution of the sea ice has been considered, the sea ice concentration, the water flux due to the melting of snow accumulated over sea ice, the salt flux due to melting/freezing and the heat flux due to snow/sea ice melting. The fields exchanged between NEMO and LIM2 are detailed in the respective documentation (Madec et al., 2016; Timmermann et al., 2005). LIM2 is a module of the NEMO code; in GLOB16, there is no need for an external coupling software to process and pass variables between the ocean and sea ice components.

- Cherchi, A., Fogli, P. G., Lovato, T., Peano, D., Iovino, D., Gualdi, S., et al.: Global mean climate and main patterns of variability in the CMCC-CM2 coupled model. Journal of Advances in Modeling Earth Systems, 11(1), 185–209. https://doi.org/10.1029/2018MS001369, 2019.
- Fogli, P.G. and D. Iovino: CMCC-CESM-NEMO: toward the new CMCC Earth System Model. CMCC Research Paper RP0248, 19 pp, 2014.
- Madec, G. and the NEMO team: "NEMO ocean engine", NEMO reference manual 3_6_STABLE, Note du Pôle de modélisation, Institut Pierre-Simon Laplace (IPSL), France, No. 27, 1288–1619, 2016.
- Timmermann, R., Goosse H., Madec G., Fichefet T., Ethe C., and V. Dulière: On the representation of high latitude processes in the orca-lim global coupled sea ice-ocean model. Ocean Modelling, 8, pp. 175-201, 2005.

111: specify what is meant by the "in-house sea ice module"? is it CICEv4.1 (line 116)? But that's not in-house.

We meant the NEMO sea ice component, so we modified the sentence.

118: specify maximum ocean depth in the 3 models

The maximum depth is indicated in Table 1.

120: "such as" - be specific. What else was used? e.g. Redi?

In NEMO, when the GM diffusion is used, an eddy induced tracer advection term is added, whose formulation depends on the slopes of iso-neutral surfaces (in z-coordinate systems, the slopes are referenced to the geopotential surfaces).

121: give GM parameter values

The eddy induced velocity coefficient is set to 1000 $m^2/s$. It is reported in Table 1.

122: "multi-category" - specify how many
Done

Table 1: suggest to also specify
- ice model dynamic and thermodynamic timesteps
- ocean-ice coupling timestep
- whether downslope transport or mixing schemes were used at low res
- whether Rayleigh was drag used in any straits at low resolution
The table now includes the ice model time steps and the coupling time step between ocean and sea ice. There are no downslope transport/mixing schemes applied in ORCA1. We specified in the text that there is strong no-slip applied in ORCA1 to reduce transport through the following straits: Gibraltar, Bhosporus, Makassar (Top), Lombok, Ombai, Timor Passage, West and East Halmahera.

133: specify JRA55-do version (1.4?)
We specified that is JRA55-do version 1.4

136: specify the interpolation methods that were used from JRA55-do to the model grid in each model
GLOB16 uses a bilinear interpolation for all variables but the wind which uses a bicubic interpolation. ORCA1 and ORCA025 use the default CESM interpolation methods based on Earth System Modeling Framework (ESMF, https://www.earthsystemmodeling.org/). In particular, state variables (temperature, pressure, etc...) are interpolated using a bilinear interpolation, fluxes using a first-order conservative remapping and vectors using a higher-order patch recovery (a second degree polynomial re-gridding method, which uses a least squares algorithm to calculate the polynomial). The runoff interpolation in all three configurations is based on the CESM default treatment which makes use of a globally conserving method which also spreads the runoff along the coast, to compute offline remapping weights.

141: is relative wind also used for stress calculation on ice?
Yes, it is also used on sea ice in all simulations. It is now specified.

145: "at the ocean surface in GLOB16" - is this literally in the top 0.8m, with no distribution to depth at all?
In GLOB16, the runoff inflow is released along the coastline. It is distributed in the surface box and spread horizontally on a set of grid points.

160: "the sea ice models used in our two systems employ different bulk salinity affecting the salt release from the sea ice to the ocean" - specify these values - how different are they? can the effect on stratification and circulation be quantified or at least estimated?
In CICE4, there is a positive flux of salt to the ocean under melting conditions, and a negative flux when sea water is freezing. In the configuration we used, a reference value of ice salinity (4psu) is used for computing the ice-ocean exchanges, although the ice salinity used in the thermodynamic calculation has differing values in the ice layers with the vertical salinity profile prescribed and fixed in time. It varies from zero at the ice top surface to $S_{max}$ (3.2 psu) at the ice bottom surface. In our version of LIM2, an equivalent salt flux is used to represent the ice/-ocean mass exchanges and is computed as the sum of freshwater flux associated with snow melting, salt rejection due to

ice formation and nearly-freshwater flux due to ice melting. Fresh water (salinity) fluxes between the ice and the ocean assume constant salinities of 6 psu. Dedicated tests should be run (for example with only ice thermodynamics activated) to quantify how the different salt release affects the upper ocean stratification and dynamics, and to isolate this effect from other differences between sea ice models. This is out of the scope of this study.

Fig 1,2 captions: specify averaging period (are annual means plotted?)
Figure 1 and Figure 2 present annual mean values. This is now specified in the captions of the two figures.

**Small errors, typos, suggestions for clearer phrasing, etc:**

7: "This paper describes the GLOB16 global ..." so GLOB16 is defined prior to first use on line 13
GLOB16 is introduced at line 9

13: access -> assess
Done

13: mesoscale activities -> resolving mesoscale processes
Done

21: effort -> efforts
Done

29: trade-off among -> trade-offs between
Done

33: Both model configurations do not resolve -> Neither resolution resolves
We modified the sentence as follows "Both resolutions lack an explicit representation of ocean mesoscale dynamics in most of the global domain"

L6: require -> requiring
We modified the sentence as follows "Simulations of the global ocean domain at this resolution still require significant computational resources, which limits the number and length of runs and the capacity to optimize the model setup. However, thanks to the ever-increasing processing and storage capabilities of the supercomputers, running global models capable of resolving mesoscale dynamics has become feasible for climate simulations. It is now necessary to assess to what extent the enhanced resolution translates into an improved ocean state."

38: access to which -> assess to what
Done

51: is -> are
Done

58: (low-, nominally 1° horizontal grid spacing), eddy-permitting (medium-, nominally 0.25°) to eddy-rich (high-, 0.0625°) resolutions -> (low-resolution, nominally 1° horizontal grid spacing), eddy-permitting (medium-resolution, nominally 0.25°) to eddy-rich (high-resolution, 0.0625°)
Grid spacing is specified afterwards. Here we only refer to low, medium and high resolution.

73: activities -> processes
Done

84: , -> , and
We have modified the sentence.

88: - the -> . The
Done

90: Outline -> An outline
Done

93: momentums -> momentum
Done

103: C grid -> a C grid
Done

105: suit -> suite
Done

107: computation cost of GLOB16 -> computational cost of the GLOB16
Done

117: refinement of meridional grid to 1/3° -> meridional refinement to 1/3°
We changed to "horizontal mesh with additional meridional refinement up to 1/3°"

123: delete "Note that the two sea ice models LIM and CICE employ different bulk salinity, affecting the salt release from the sea ice into the ocean." - repeated in line 160
Done

141: JRA55 -> JRA55-do
Done

148: GLOB16 grid -> The GLOB16 grid
Done

154: "in coupled runs" is unclear. I suggest "in ORCA1 and ORCA025" (assuming this is what was meant)
Yes, we changed the sentence as suggested

167: that warms -> warming
Done

169: staying anyway -> but staying
Done

170: specify whether the 0.1°C cooling is over 1 cycle or 6
Done

175: over -> out of
Done

187: 2c -> 1b
Done

189: what found -> what was found
Done

fig 1: swap b & c in caption
Done

205: largely -> greatly
Done

212: resolutions -> resolution
Done

229: evolution, the -> evolution, and the
Done

244: in 2018 -> by 2018?
Done

258: activities -> activity
Done

259: GLOB16 bias -> GLOB16
Done

261: define "WBC" acronym on first use
Done

262: differences are also in -> improvements are also seen in
Done

265: regions -> regions than ORCA1 and ORCA025
We added "compared to ORCA1 and ORCA025".

268: experiments with a -> experiments due to a
Done

275: insert last sentence of this paragraph here as the 2nd sentence
Done

288: The mixed -> The winter mixed
Done

290: Ross and Weddell convection is cut off in figs 5,6

We changed figures 5 [*figure 6 in the revised manuscript*] and 6 [*figure 7 in the revised manuscript*]. They include all the southern hemisphere to better show the Weddell and Ross Seas.

298: remove "slightly" - this is quite a big difference
Done

301 high-latitude -> northern high-latitude
Done

305: horizontal -> zonal?
We mean a wider extension both in meridional and zonal direction. We did not modify the word.

306: remove "caveats"
Done

314: NH Sept MLD not interpretable with this colormap
To better adapt the colorbar, figure 5 [*figure 6 in the revised manuscript*] shows the MLD averaged over March (in the Northern Hemisphere) and September (in the Southern Hemisphere), figure 6 [*figure 7 in the revised manuscript*] shows the MLD averaged over September (in the Northern Hemisphere) and March (in the Southern Hemisphere).

329, table 1: were downslope transport or mixing schemes used at low res?
ORCA1 does not use downslope transport or mixing schemes to be indicated in the table.

350: the observation one -> the spread of observations
Done

360: (Fig. 8), zoomed in the key dynamical regions (Fig. 9 and 10), -> (Fig. 8), and zoomed in to the key dynamical regions (Fig. 9 and 10). The
Done

404: passed -> past
Done

404: follows closely -> closely follows
Done

407: to OSCAR -> to the OSCAR
Done

408: remove "in amplitude"
Done

408: The decaying eastward along the Kuroshio extension and magnitude match the observed one -> The GLOB16 Kuroshio extension magnitude and its eastward decay match observations
Done

409: the 170°E longitude -> 170°E
Done

410: toward 145°E, to rapidly decay westwards. -> until 145°E, but decays too rapidly further east.
Done

Fig 9: fix panel numbering (a-d happens twice)
Done

464, 474, 475, fig 11: stated transports are double what is shown in fig 11 - is the text incorrect or is the contour interval 2 Sv, not 1 Sv as stated in the caption for fig 11?
The contour interval is 2 Sv as now stated in the caption.

480: "~2 Sv in density space (~ 6 Sv below 3000 m in depth space)." Are these back to front? Looks like about 6 Sv in fig 11 (or 12 Sv if the contour interval is 2 Sv)
The abyssal cell reaches ~12 Sv. Correction done.

467: "A portion" - quantify this
We indicated in the text that about 4 Sv move northward across 30°S in the GLOB16 simulation.

468: we do present -> we present
Done

473: 55°N -> 65°N?
We changed the sentence as follows "In GLOB16, the NADW starts to sink north of 45°N with the maximum transport located at 55°N and the largest densification north of 60°N".

Fig 11: plot contours beyond current range (both positive and negative) - seem to be missing some in the far south, unless the extrema are very flat. Also state in caption how colour is related to overturning direction
The plots cover the entire domain in the southern boundary. In the caption, it is now specified that positive (negative) stream function indicates strength in the clockwise (counter-clockwise) direction.

497: till -> until
Done

Fig 12 caption: Swap "dashed" and "solid" and state that ORCA1 is also plotted.
Done

512: two decades -> decade?
Done

513: can't see low in 2009 in either obs record in Fig 13. "observed in 2009 and 2010" -> "observed at 26.5°N in 2005, 2010, 2011 and 2013"?
Thank you, corrected.

518, 528, fig 13: why not plot the other 2 models as well?
We decided to have the mean values of ORCA1 and ORCA025 meridional overturning and not show the time series since we want to keep the focus of the paper on GLOB16 results.
The time variability is similar among simulations as shown in the plots below:

[Figure]

Figure RC2_6. time evolution of monthly mean AMOC transports, defined as the maximum value of the global overturning stream function in GLOB16 (blue line) ORCA025 (green line) and ORCA1 (orange line) computed (a) across 26.5°N and compared to RAPID estimates, and (b) 34°S compared to SAMBA record.

Fig 13 caption: state that the time scale is compressed prior to 2000.
Done

539: 15°N -> 18°N?
Done

540, 541: swap 10 - 20°N and 20-30°N (or is fig14a key wrong?)
Done

542: MHT peaks around 24°N -> MHT peaks around 24°N but is not well constrained given the error bars
we have added uncertainties in the sentence

542-589: MHT -> AMHT
Corrected throughout the manuscript

548: remove "The strongest heat transport is found in the eddy-rich ocean." - repeated below
Done

550: GLOB16 tracks the ECMWF estimates and compares well with TF08 -> GLOB16 tracks the ECMWF estimates and compares well with TF08 except north of 40°N
Thank you, corrected.

552: Hirschi et al. 2019 -> Hirschi et al. 2020?
Done

556: 18°N -> 15°N ?
the latitude is correct now

565: slightly exceeds the total MHT between the equator and 15°N -> slightly exceeds the total
MHT between the equator and 15°N and south of 20°S
Done

568: gyre one increases to level off -> gyre component increases to level off at total AMHT
Done

574: remove "not shown for ORCA025 575 and ORCA1" - this difference is visible in fig 14a.
we point to Figure 14a now

578: misrepresent -> misrepresents
Done

579: 2010, 2011, 2013 and 2017 minima -> 2010, 2011, 2013, 2017 and 2019 minima?
The time series covers the integration time from 1958 to 2018. The 2019 minimum is not presented.

Fig 14b: why not include the other 2 models?
As for figure 13 [*figure 14 in the revised manuscript*], we decided not to present the time series of
the meridional heat transport since we want to keep the focus of the paper on GLOB16 results. The
mean values for ORCA1 and ORCA025 simulations are indicated in the text. The time variability
is similar among simulations as shown in the plots below:

[Figure]

Figure RC2_7. Times series of the monthly-mean total AMHT in GLOB16 (blue line), ORCA025 (green line) and
ORCA1 (orange line) across 26.5°, compared to the RAPID record (magenta).

608: negative -> positive
We changed the plot, so the negative transport indicates water from the Pacific to the Indian ocean
now

617, table 1: was Rayleigh drag used in the straits at low resolution?
No, the Rayleigh friction is not used in the ORCA1 simulation to improve the transport through
straits.

629: plot Donohue et al value in fig 15c

The value of cDrake total transport (173.3 ± 10.7 Sv, Donohue et al., 2016) is indicated in the text. It combines the mean baroclinic transport (127.7±8.1 Sv from Chidichimo et al. 2014) with the mean depth-independent transport estimate of 45.6±8.9 Sv. We decided not to include cDrake in the figure 15c [*figure 16c in the revised manuscript*] since it is largely higher than all other observation-based products. With an error estimate of 10.7 Sv, the cDrake minimum estimate is of 163Sv, above mean model values and other estimates.

631: below the most recent estimates (Xu et al., 2020) -> below the most recent estimates (Xu et al., 2020) and some eddy-rich models (Kiss et al., 2020)
Done

Fig 15 caption: (b) the Indonesian Throughflow -> (b) the southward Indonesian Throughflow
We modified the plot. Negative values are from the Pacific to the Indian ocean now

Fig 15 caption: Cite sources for obs in a & b
Done

671: provide -> provides
Done

679: Indian to the Pacific -> Pacific to Indian
Done

683: (weaker than observed values) -> (weaker than observed values and some other eddy-rich models)
We added this line into session 3.3.2. Since the manuscript does not aim to provide a direct comparison with other model simulations, we prefer not to add it in the conclusion.

---

## Author Response (AR2)

Dear Editor

We would like to thank you for your latest suggestions. We corrected the manuscript accordingly, along with some typos and corrections to the reference list. All authors agree with the modifications.

For the authors,
Doroteaciro Iovino